# Replay-and-Forget-Free Graph Class-Incremental Learning: A Task Profiling and Prompting Approach

**Chaoxi Niu[1], Guansong Pang[2]\*, Ling Chen[1], Bing Liu[3]**

[1] AAII, University of Technology Sydney, Australia

[2] School of Computing and Information Systems, Singapore Management University, Singapore

[3] Department of Computer Science, University of Illinois at Chicago, USA

chaoxi.niu@student.uts.edu.au,   gspang@smu.edu.sg

ling.chen@uts.edu.au,   liub@uic.edu

## Abstract

Class-incremental learning (CIL) aims to continually learn a sequence of tasks, with each task consisting of a set of unique classes. Graph CIL (GCIL) follows the same setting but needs to deal with graph tasks (*e.g.*, node classification in a graph). The key characteristic of CIL lies in the absence of task identifiers (IDs) during inference, which causes a significant challenge in separating classes from different tasks (*i.e.*, *inter-task class separation*). Being able to accurately predict the task IDs can help address this issue, but it is a challenging problem. In this paper, we show theoretically that accurate task ID prediction on graph data can be achieved by a Laplacian smoothing-based graph task profiling approach, in which each graph task is modeled by a task prototype based on Laplacian smoothing over the graph. It guarantees that the task prototypes of the same graph task are nearly the same with a large smoothing step, while those of different tasks are distinct due to differences in graph structure and node attributes. Further, to avoid the *catastrophic forgetting* of the knowledge learned in previous graph tasks, we propose a novel *graph prompting* approach for GCIL which learns a small discriminative graph prompt for each task, essentially resulting in a separate classification model for each task. The prompt learning requires the training of a single graph neural network (GNN) only once on the first task, and no data replay is required thereafter, thereby obtaining a GCIL model being both **replay-free** and **forget-free**. Extensive experiments on four GCIL benchmarks show that i) our task prototype-based method can achieve 100% task ID prediction accuracy on all four datasets, ii) our GCIL model significantly outperforms state-of-the-art competing methods by at least 18% in average CIL accuracy, and iii) our model is fully free of forgetting on the four datasets. Code is available at https://github.com/mala-lab/TPP.

## 1 Introduction

Graph continual learning (GCL) [5, 28, 39, 43] aims to continually learn a model that not only accommodates the new emerging graph data but also maintains the learned knowledge of previous graph tasks, with each graph task comprising nodes from a set of unique classes in a graph. Due to privacy concerns and the hardware limitations in storage and computation, GCL assumes that the data of previous graph tasks is not accessible when learning new graph tasks. This leads to *catastrophic forgetting* of the learned knowledge, *i.e.*, degraded classification accuracy on previous tasks due to model updating on new tasks.

---

\*Corresponding author: G. Pang (gspang@smu.edu.sg)

38th Conference on Neural Information Processing Systems (NeurIPS 2024).

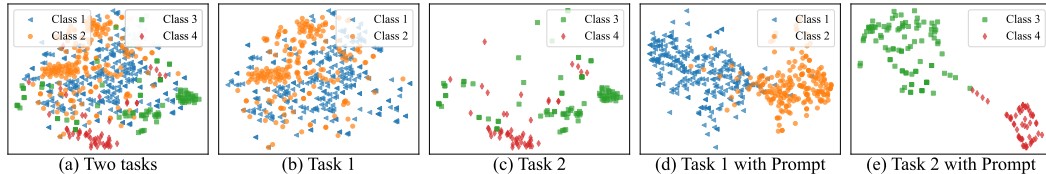

Figure 1: **(a)** Classification space of two graph tasks when no task ID is provided. The classification space is split into two separate spaces in Task 1 in **(b)** and Task 2 in **(c)** when the task ID can be accurately predicted. This helps alleviate the inter-task class separation issue. To mitigate catastrophic forgetting, we learn a graph prompt for each task that absorbs task-specific discriminative information for better class separation within each task, as shown in **(d)** and **(e)** respectively. This essentially results in a separate classification model for each task, achieving fully forget-free GCIL models.

*Graph class-incremental learning* (GCIL) is one key setting of GCL, in which task identifiers (IDs) are not provided during inference. *Graph task-incremental learning* (GTIL) is another GCL setting where the task ID is given for each test sample. As a result, a set of separate classifiers can be learned for different graph tasks in GTIL and the task-specific classifier can be used for each test sample. Compared to GTIL, the absence of task IDs in GCIL presents an additional challenge, known as *inter-task class separation* [10, 11, 14], *i.e.*, the class separation in one graph task is obstructed by the presence of classes from other tasks. Consequently, the classification performance in GCIL is typically far below that in GTIL [15, 17, 20, 41, 42, 44]. This paper focuses on the GCIL setting – a more compelling GCL problem – aiming to bridge the performance gap between GCIL and GTIL.

Existing GCL methods often alleviate the catastrophic forgetting through preserving important model parameters of previous tasks [15], continually expanding model parameters for new tasks [37, 40], or augmenting with a memory module for data replay [17, 20, 20, 41, 42, 44]. However, their ability to handle the inter-task class separation is limited, leading to poor GCIL classification accuracy, especially the accuracy of the previous graph tasks. Thus, existing GCL methods typically show substantially higher forgetting in GCIL than in GTIL.

To address these issues, we introduce a novel GCIL approach, namely **Task Profiling and Prompting (TPP)**. In particular, we reveal for the first time theoretically and empirically that the task ID of a test sample can be accurately predicted by using a Laplacian smoothing-based method to profile each graph task with a prototypical embedding. With the existence of edges between nodes, this task profiling method guarantees that the task prototypes of the same graph task are nearly the same with a large smoothing step, while those of different tasks are distinct due to differences in graph structure and node attributes. High task ID prediction accuracy helps confine the classification space of the test samples to the classes of the predicted task (*e.g.*, Task 1 or 2 in Fig. 1b,c) instead of all the learned tasks (*e.g.*, both tasks as in Fig. 1a), eliminating the inter-task class separation issue. There have been some studies on task ID prediction [10, 11, 14], but they are designed for Euclidean data that are i.i.d. (independent and identically distributed). As a result, they fail to leverage the graph structure and node attributes in the non-Euclidean graph data and are not suited for graph task ID prediction.

To address the catastrophic forgetting problem, we further propose a novel graph prompting approach for GCIL. Specifically, we optimize a single, small learnable prompt using a simple frozen pre-trained graph neural network (GNN) to capture the task-specific knowledge for each graph task during training. Despite being small, the task-specific knowledge learned in the prompts can ensure the intra-task class separation, as shown in Fig. 1d,e. At test time, given a test graph, the task prototype constructed with its structure and node attributes is utilized for task ID prediction, and the graph prompt of the predicted task is incorporated into the test graph for classification with the GNN. Since the graph prompts are learned task by task, *no data replay* is required in our TPP model. Further, the graph prompts are task-specific, so we essentially have a separate classification model for each graph task, *i.e.*, no continual model updating, completely avoiding the forgetting problem (*i.e.*, *forget-free*).

Overall, this work makes the following main contributions. **(1)**: We propose a novel graph task profiling and prompting (TPP) approach, which is the first replay- and forget-free GCIL approach. **(2)**: We reveal theoretically that a simple Laplacian smoothing-based graph task profiling approach can achieve accurate graph task ID prediction. To the best of our knowledge, it is the first work that leverages the non-Euclidean properties of graph data to enable graph task ID prediction. It achieves

100% prediction accuracy across all four datasets used, eliminating the inter-task class separation issue in GCIL. **(3)**: We further introduce a novel graph prompting approach that learns a small prompt for each task using a frozen pre-trained GNN, without any data replay involved. With the support of our accurate task ID prediction, the graph prompts result in a separate classification model for each task, resulting in the very first GCIL model being both replay-free and forget-free. **(4)**: Extensive experiments on four GCIL benchmarks show that our TPP model significantly outperforms state-of-the-art competing methods by at least 18% in average CIL accuracy while being fully forget-free. It even exceeds the joint training on all tasks simultaneously by a large margin.

## 2 Related Work

**Graph Continual Learning.** Various methods have been proposed for GCL [7,15,21,22,24,25,31,37, 38,41,42,44] and can be divided into three categories, *i.e.*, regularization-based, parameter isolation-based, and data replay-based methods. Regularization-based methods typically preserve parameters that are important to the previous tasks when learning new tasks via additional regularization terms. For example, TWP [15] preserves the important parameters in the topological aggregation and loss minimization for previous tasks via regularization terms. Parameter isolation-based methods maintain the performance on previous tasks by continually introducing new parameters for new tasks such as [37] proposes to continually expand model parameters to learn new emerging graph patterns. Differently, replay-based methods [17,20,41,42,44] employ an additional memory buffer to store the information of previous tasks and replay them when learning new tasks. The ways to construct the memory buffer play a vital role in replay-based methods. Despite they have shown good performance in alleviating the forgetting problem, inter-class separation is still a significant challenge to these methods, especially for the CIL setting where the task IDs are not provided when testing.

To improve the CIL performance, an emerging research direction focuses on performing task ID prediction during testing. For example, CCG [1] utilizes a separate network for task identification. HyperNet [30] and PR-Ent [8] use the entropy to predict the task of the test sample. More recently, [10] proves that OOD detection is the key to task ID prediction and proposed an identification method based on an OOD detector. TPL [14] further improves it by exploiting the information available in CIL for better task identification. Since these methods were designed for Euclidean data, they are not suited for GCIL. Following this line, we propose a task ID prediction method specifically for GCIL in this paper. Different from previous methods that rely on additional networks or OOD detectors, the proposed task identification is accomplished by using a Laplacian smoothing-based method to profile each graph task with a prototypical embedding. Despite its simplicity, this graph task profiling method can achieve accurate task ID prediction with theoretical support.

**Prompt Learning.** Originating from natural language processing, prompt learning aims to facilitate the adaptation of frozen large-scale pre-trained models to various downstream tasks by introducing learnable prompts [16]. In other words, prompt learning designs task-specific prompts to instruct the pre-trained models to perform downstream tasks conditionally. The prompts capture the knowledge of the corresponding tasks and enhance the compatibility between inputs and pre-trained models. Recently, prompt-based graph learning methods have also been proposed [27], which aims to unify multiple graph tasks [26] or improve the transferability of graph models [4]. Due to the ability to leverage the strong representative capacity of the pre-trained model and learn the knowledge of tasks in prompts, many prompting-based continual learning methods have been proposed [32–34] and achieved remarkable success without employing replaying memory or regularization terms. Despite that, no work is done on prompt learning for GCIL. The main challenge is the lack of pre-trained GNN models for all tasks in GCIL and the absence of task IDs to retrieve corresponding prompts during testing. In this work, we show that effective graph prompts can be learned for different tasks using a GNN backbone trained based on the first task with graph contrastive learning [36,45], and our task ID prediction method and the graph prompts can be synthesized to address the GCIL problem.

## 3 Methodology

### 3.1 The GCIL Problem

Formally, GCL can be formulated as learning a model on a sequence of connected graphs (tasks) $\{\mathcal{G}^1, \ldots, \mathcal{G}^T\}$ where $T$ is the number of tasks. Each $\mathcal{G}^t = (A^t, X^t)$ is a newly emerging graph,

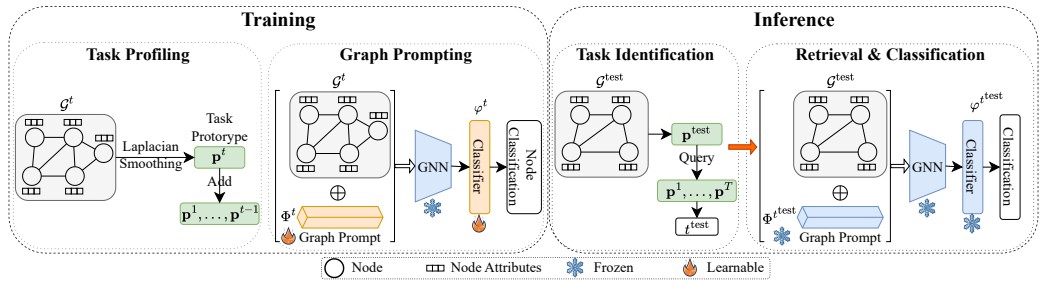

Figure 2: Overview of the proposed TPP approach. During training, for each graph task $t$, the task prototype $\mathbf{p}^t$ is generated by applying Laplacian smoothing on the graph $\mathcal{G}^t$ and added to $\mathcal{P} = \{\mathbf{p}^1, \ldots, \mathbf{p}^{t-1}\}$. At the same time, the graph prompt $\Phi^t$ and the classification head $\varphi^t$ for this task are optimized on $\mathcal{G}^t$ through a frozen pre-trained GNN. During inference, the task ID of the test graph is first inferred (*i.e.*, task identification). Then, the graph prompt and the classifier of the predicted task are retrieved to perform the node classification in GCIL. The GNN is trained on $\mathcal{G}^1$ and remains frozen for subsequent tasks.

where $A^t$ denotes the relations between $N$ nodes of the current/new task, $X^t \in \mathbb{R}^{N \times F}$ represents the node attributes with dimensionality of $F$, and the labels of nodes can be denoted as $Y^t$. Each task contains a unique set of classes in a graph, i.e., $\{Y^t \cap Y^j = \varnothing | t \neq j\}$. When learning task $t$, the model trained from previous tasks only has access to the current task data $\mathcal{G}^t$. The goal is to accommodate the model to current graph $\mathcal{G}^t$ while maintaining the classification performance on the previous graphs $\{\mathcal{G}^1, \ldots, \mathcal{G}^{t-1}\}$. In GCIL, the task IDs are not available during inference. Thus, assuming that each task has $C$ classes, after learning all tasks, a GCIL model is required to classify a test instance into one of all the $T \times C$ classes.

## 3.2 Overview of The Proposed TPP Approach

Inspired by prior studies [10, 11], we decompose the class probability of a test sample $\mathbf{x}^{\text{test}}$ belonging to the $j$-th class in task $t$ in GCIL into two parts :

$$H(y_j^t | \mathbf{x}^{\text{test}}) = H(y_j^t | \mathbf{x}^{\text{test}}, t) H(t | \mathbf{x}^{\text{test}}), \tag{1}$$

where $H(t | \mathbf{x}^{\text{test}})$ represents the task ID prediction probability of task $t$ and $H(y_j^t | \mathbf{x}^{\text{test}}, t)$ denotes the prediction within the task $t$. This indicates that accurate GCIL classification accuracy can be achieved when both accurate task ID prediction and intra-task class classification are achieved.

To this end, in this paper, we propose the Task Profiling and Prompting (**TPP**) approach for GCIL. As shown in Fig. 2, a novel Laplacian smoothing-based task profiling approach is first devised in TPP for graph task ID prediction, which can well guarantee the task prediction accuracy as we will demonstrate theoretically below. Moreover, to obtain accurate intra-task class classification within the identified task, a novel graph prompting approach is further proposed to learn a small prompt for each task using a frozen GNN pre-trained on the first graph task. By learning and storing task knowledge separately, there is no knowledge interference between tasks during training, resulting in a model being both replay-free and forget-free. During inference, given a test sample, TPP first performs the task ID prediction and then retrieves the corresponding task graph prompt to concatenate with the sample for the GCIL classification. Below we introduce the TPP approach in detail.

## 3.3 Laplacian Smoothing-based Task Profiling for Graph Task ID Prediction

To leverage the graph structure and node attribute information, we propose to use a Laplacian smoothing approach to generate a prototypical embedding for each graph task for task ID prediction. Specifically, for the task $t$ with graph data $\mathcal{G}^t = (A^t, X^t)$, we construct a task prototype $\mathbf{p}^t$ for this task based on the train set denoted as $\{x_i | i \in \mathcal{V}_{\text{train}}^t\}$, where $\mathcal{V}_{\text{train}}^t$ is the train set of $\mathcal{G}^t$. Given $\mathcal{G}^t$, the Laplacian smoothing is first applied on the graph $\mathcal{G}_t$ to obtain the smoothed node embeddings $Z^t$:

$$Z^t = (I - (\hat{D}^t)^{-\frac{1}{2}} \hat{L}^t (\hat{D}^t)^{-\frac{1}{2}})^s X^t, \tag{2}$$

where $s$ denotes the number of Laplacian smoothing steps, $I$ is an identity matrix, and $\hat{L}^t$ is the graph Laplacian [3] matrix of $\hat{A}^t = A^t + I$ (*i.e.*, $\hat{L}^t = \hat{D}^t - \hat{A}^t$ with $\hat{D}^t$ being the diagonal degree matrix

of $\hat{A}^t$ and $\hat{D}_{ii}^t = \sum_j \hat{a}_{ij}$). Then, the task prototype $\mathbf{p}^t$ is constructed by averaging the smoothed embeddings of train nodes:

$$\mathbf{p}^t = \frac{1}{|\mathcal{V}_{\text{train}}^t|} \sum_{i \in \mathcal{V}_{\text{train}}^t} \mathbf{z}_i^t (\hat{D}_{ii}^t)^{-\frac{1}{2}} . \tag{3}$$

Similarly, the task prototypes for all tasks can be separately constructed and stored, denoted as $\mathcal{P} = \{\mathbf{p}^1, \dots, \mathbf{p}^T\}$. Given a test graph $\mathcal{G}^{\text{test}}$ at testing time, we predict the task ID of $\mathcal{G}^{\text{test}}$ by querying the task prototype pool $\mathcal{P}$. Specifically, the task prototype of $\mathcal{G}^{\text{test}}$ is obtained with the set of test nodes in a similar way as on training graphs via Eq. (2) and Eq. (3), *i.e.*,

$$\mathbf{p}^{\text{test}} = \frac{1}{|\mathcal{V}^{\text{test}}|} \sum_{i \in \mathcal{V}^{\text{test}}} \mathbf{z}_i^{\text{test}} (\hat{D}_{ii}^{\text{test}})^{-\frac{1}{2}} , \tag{4}$$

where $\mathcal{V}^{\text{test}}$ denotes the set of nodes to be classified in $\mathcal{G}^{\text{test}}$ and $\mathbf{z}_i^{\text{test}}$ is the smoothed embedding of the test node $i$ after $s$-step Laplacian smoothing. Then, we query the task prototype pool $\mathcal{P}$ with the test prototype $\mathbf{p}^{\text{test}}$ and return the task ID whose task prototype is most similar to $\mathbf{p}^{\text{test}}$:

$$t^{\text{test}} = \arg\min(d(\mathbf{p}^{\text{test}}, \mathbf{p}^1), \dots, d(\mathbf{p}^{\text{test}}, \mathbf{p}^T))) , \tag{5}$$

where $d(\cdot)$ represents an Euclidean distance function and $t^{\text{test}}$ is the predicted task ID of $\mathcal{G}^{\text{test}}$.

As discussed in Sec. 3.2, more accurate task ID prediction leads to better classification performance for GCIL. Below we show theoretically that the task ID of the test graphs can be accurately predicted with our simple Laplacian smoothing-based task profiling approach.

**Theorem 1.** *If graphs for all tasks are not isolated and the test graph $\mathcal{G}^{test}$ comes from the task $t$, i.e., $\mathcal{G}^{test}$ and $\mathcal{G}^t$ have the same set of classes, then the distance between $\mathbf{p}^{test}$ and $\mathbf{p}^t$ approaches to zero with a sufficiently large number of Laplacian smoothing steps $s$:*

$$\lim_{s \to +\infty} d(\mathbf{p}^{test}, \mathbf{p}^t) = 0 . \tag{6}$$

**Theorem 2.** *Suppose the test graph comes from task $t$, and let $\mathbf{e}$ and $\epsilon$ be the differences in node degrees and node attributes between two different tasks $t$ and $j$ respectively, which are defined by $(\hat{D}^j)^{\frac{1}{2}} = (\hat{D}^t)^{\frac{1}{2}} + \text{Diag}(\mathbf{e})$ and $X^j = X^t + \epsilon$. Then the distance between the task prototypes of task $t$ and $j$ obtained with large steps of Laplacian smoothing can be explicitly calculated as:*

$$d(\mathbf{p}^{test}, \mathbf{p}^j) = \|(\mathbf{e}_N^t)^T \epsilon + (\mathbf{e})^T X^j\|_2 , \tag{7}$$

*where $\mathbf{e}_N^t = (\hat{D}^t)^{\frac{1}{2}}[1, 1, \dots, 1]^T$ is the $N$-th eigenvector of task $t$ and $(\mathbf{e}_N^t)^T$ denotes its transpose.*

The two theorems indicate that i) if the test graph belongs to task $t$, with a large $s$, the distance between $\mathbf{p}^{\text{test}}$ and $\mathbf{p}^t$ would become zero with the proposed Laplacian smoothing and prototype construction method (Theorem 1); and ii) for graphs from different tasks, since they contain different set of classes, they have large differences in graph structure and node attributes, which can lead to a large distance between task prototypes $\mathbf{p}^t$ and $\mathbf{p}^j$ (Theorem 2), thereby having the following inequality hold if $\mathcal{G}^{\text{test}}$ comes from task $t$:

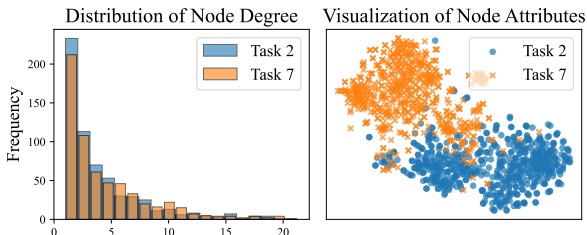

Figure 3: The differences between two graphs in structure and node attributes.

$$d(\mathbf{p}^{\text{test}}, \mathbf{p}^t) < d(\mathbf{p}^{\text{test}}, \mathbf{p}^j) . \ \forall j \neq t . \tag{8}$$

Without loss of generality, we empirically investigate the differences between two randomly chosen graph tasks of the CorFull dataset in Fig. 3. We can see that the two graphs of the tasks have a rather large difference in both graph structure and node attributes. The larger the differences in $\mathbf{e}$ and $\epsilon$ of the two graphs, the larger the gap is between $d(\mathbf{p}^{\text{test}}, \mathbf{p}^t)$ and $d(\mathbf{p}^{\text{test}}, \mathbf{p}^j)$. As a result, the task of the test graph can be predicted accurately with Eq. (5). In the experimental section, we empirically evaluate the proposed task ID prediction and report its accuracy on different datasets.

## 3.4 Graph Prompt Learning for GCIL

Instead of utilizing regularization or replaying memory as in existing GCL methods, TPP aims to learn a task-specific prompt for each graph task. The information of each graph task can be explicitly modeled and stored in a separate task-specific graph prompt, with the GNN backbone being frozen. This effectively avoids the forgetting of knowledge of any previous tasks and the interference between tasks. To this end, the graph prompt in TPP is designed as a set of learnable tokens that can be incorporated into the feature space of the graph data for each task. Specifically, for a task $t$, the graph prompt can be represented as $\Phi^t = [\phi_1^t, \dots, \phi_k^t]^T \in \mathbb{R}^{k \times F}$ where $k$ is the number of vector-based tokens $\phi^i$. For each node in $\mathcal{G}^t$, the node attribute is augmented by the weighted combination of these tokens, with the weights obtained from $k$ learnable linear projections:

$$\bar{\mathbf{x}}_i^t = \mathbf{x}_i^t + \sum_j^k \alpha_j \phi_j^t, \quad \alpha_j = \frac{e^{(\mathbf{w}_j)^T \mathbf{x}_i^t}}{\sum_l^k e^{(\mathbf{w}_l)^T \mathbf{x}_i^t}}, \tag{9}$$

where $\alpha_j$ denotes the importance score of the token $\phi^j$ in the prompt and $\mathbf{w}_j$ is a learnable projection. For convenience, we denote the graph modified by the graph prompt as $\bar{\mathcal{G}}^t = (A^t, X^t + \Phi^t)$. Then, $\bar{\mathcal{G}}^t$ is fed into a frozen pre-trained GNN model $f(\cdot)$ to obtain the embeddings for classification. In TPP, we employ a single-layer MLP as the classifier attached to the GNN, denoting $\varphi^t$ for task $t$. The node classification at task $t$ can be formulated as:

$$\hat{Y}^t = \varphi^t(f(A^t, X^t + \Phi^t)). \tag{10}$$

Therefore, the graph prompt and the MLP-based classification head are optimized by minimizing a node classification loss:

$$\min_{\Phi^t, \varphi^t} \frac{1}{|\mathcal{V}_{\text{train}}^t|} \sum_{i \in \mathcal{V}_{\text{train}}^t} \ell_{\text{CE}}(\hat{y}_i^t, y_i^t), \tag{11}$$

where $\mathcal{V}_{\text{train}}^t$ is the train set of $\mathcal{G}^t$, $y_i^t$ is the label of node $i$, $\hat{y}_i^t \in \hat{Y}^t$ is the predicted label, and $\ell_{\text{CE}(\cdot)}$ is a cross-entropy loss. By minimizing Eq. (9), the graph prompt and the classifier are learned to leverage the generic, cross-task knowledge in the frozen GNN $f(\cdot)$ for the task $t$. Meanwhile, $\Phi^t$ and $\varphi^t$ learn specific knowledge for the task $t$. This essentially results in a separate classification model for each task, and no data replay is required for all tasks. As a result, TPP is fully free of catastrophic forgetting for GCIL. An alternative approach to overcoming the forgetting is to learn a separate GNN model for each task. However, this would introduce heavy burdens on optimization and storage with the increasing number of tasks. By contrast, the proposed graph prompting only introduces minimal parameters for each task as the prompts are very small.

## 3.5 Training and Inference in TPP

**Training.** The training of TPP can be divided into two parts. First, for each task $\mathcal{G}^t$, the prototypical embedding $\mathbf{p}^t$ is generated based on Laplacian smoothing and stored in $\mathcal{P}$ for task ID prediction. Then, the information of $\mathcal{G}^t$ is explicitly modeled and stored with the proposed graph prompt learning, *i.e.*, Eq. (11). For the GNN backbone $f(\cdot)$ in graph prompt learning, we propose to learn it based on the first task $\mathcal{G}^1 = (A^1, X^1)$ via graph contrastive learning due to its ability to obtain transferable models [36, 45] across graphs (see Appendix B). Despite being only learned on $\mathcal{G}^1$, $f(\cdot)$ can effectively adapt to all subsequent tasks with graph prompts. Overall, after learning all tasks in $\{\mathcal{G}^1, \dots, \mathcal{G}^T\}$, the task profiles and task-specific information are explicitly modeled in $\mathcal{P} = \{\mathbf{p}^1, \dots, \mathbf{p}^T\}$, $\{\Phi^1, \dots, \Phi^T\}$ and $\{\varphi^1, \dots, \varphi^T\}$.

**Inference.** Given the test graph $\mathcal{G}^{\text{test}}$, the task prototype $\mathbf{p}^{\text{test}}$ is constructed with Eq. (4) and then used to obtain the task ID $t^{\text{test}}$ by querying $\mathcal{P} = \{\mathbf{p}^1, \dots, \mathbf{p}^T\}$, *i.e.*, via Eq. (5). Finally, the test graph $\mathcal{G}^{\text{test}}$ is augmented with the corresponding graph prompt $\Phi^{t^{\text{test}}}$ and fed into the GNN and the classification head constructed with $f(\cdot)$ and $\varphi^{t^{\text{test}}}$ respectively to get the node classification results. Formally, the inference can be formulated as:

$$\begin{cases} t^{\text{test}} = \arg\min(d(\mathbf{p}^{\text{test}}, \mathbf{p}^1), \dots, d(\mathbf{p}^{\text{test}}, \mathbf{p}^T))), \\ Y^{\text{test}} = \varphi^{t^{\text{test}}}(f(A^{\text{test}}, X^{\text{test}} + \Phi^{t^{\text{test}}})). \end{cases} \tag{12}$$

The algorithms of the training and inference of TPP are provided in Appendix C.

Table 1: Results (mean±std) under the GCIL setting on four large datasets. The best performance on each dataset is boldfaced. "↑" denotes the higher value represents better performance. Oracle Model can get access to the data of all tasks and task IDs, *i.e.*, it obtains the upper bound performance. "✓" in Data Replay indicates the use of data replay in the model, and × denotes no data replay involved.

| Methods | Data Replay | CoraFull | | Arxiv | | Reddit | | Products | |
|---|---|---|---|---|---|---|---|---|---|
| | | AA/%↑ | AF/%↑ | AA/%↑ | AF/%↑ | AA/%↑ | AF/%↑ | AA/%↑ | AF/%↑ |
| Fine-tune | × | 3.5±0.5 | -95.2±0.5 | 4.9±0.0 | -89.7±0.4 | 5.9±1.2 | -97.9±3.3 | 7.6±0.7 | -88.7±0.8 |
| Joint | × | 81.2±0.4 | - | 51.3±0.5 | - | 97.1±0.1 | - | 71.5±0.1 | - |
| EWC | × | 52.6±8.2 | -38.5±12.1 | 8.5±1.0 | -69.5±8.0 | 10.3±11.6 | -33.2±26.1 | 23.8±3.8 | -21.7±7.5 |
| MAS | × | 6.5±1.5 | -92.3±1.5 | 4.8±0.4 | -72.2±4.1 | 9.2±14.5 | -23.1±28.2 | 16.7±4.8 | -57.0±31.9 |
| GEM | × | 8.4±1.1 | -88.4±1.4 | 4.9±0.0 | -89.8±0.3 | 11.5±5.5 | -92.4±5.9 | 4.5±1.3 | -94.7±0.4 |
| LwF | × | 33.4±1.6 | -59.6±2.2 | 9.9±12.1 | -43.6±11.9 | 86.6±1.1 | -9.2±1.1 | 48.2±1.6 | -18.6±1.6 |
| TWP | × | 62.6±2.2 | -30.6±4.3 | 6.7±1.5 | -50.6±13.2 | 8.0±5.2 | -18.8±9.0 | 14.1±4.0 | -11.4±2.0 |
| ERGNN | ✓ | 34.5±4.4 | -61.6±4.3 | 21.5±5.4 | -70.0±5.5 | 82.7±0.4 | -17.3±0.4 | 48.3±1.2 | -45.7±1.3 |
| SSM-uniform | ✓ | 73.0±0.3 | -14.8±0.5 | 47.1±0.5 | -11.7±1.5 | 94.3±0.1 | -1.4±0.1 | 62.0±1.6 | -9.9±1.3 |
| SSM-degree | ✓ | 75.4±0.1 | -9.7±0.0 | 48.3±0.5 | -10.7±0.3 | 94.4±0.0 | -1.3±0.0 | 63.3±0.1 | -9.6±0.3 |
| SEM-curvature | ✓ | 77.7±0.8 | -10.0±1.2 | 49.9±0.6 | -8.4±1.3 | 96.3±0.1 | -0.6±0.1 | 65.1±1.0 | -9.5±0.8 |
| CaT | ✓ | 80.4±0.5 | -5.3±0.4 | 48.2±0.4 | -12.6±0.7 | 97.3±0.1 | -0.4±0.0 | 70.3±0.9 | -4.5±0.8 |
| DeLoMe | ✓ | 81.0±0.2 | -3.3±0.3 | 50.6±0.3 | 5.1±0.4 | 97.4±0.1 | -0.1±0.1 | 67.5±0.7 | -17.3±0.3 |
| OODCIL | ✓ | 71.3±0.5 | -1.1±0.1 | 19.3±1.4 | -1.0±0.4 | 79.3±0.8 | -0.1±0.0 | 41.6±0.9 | -1.6±0.4 |
| TPP (Ours) | × | **93.4±0.4** | **0.0±0.0** | **85.4±0.1** | **0.0±0.0** | **99.5±0.0** | **0.0±0.0** | **94.0±0.5** | **0.0±0.0** |
| Oracle Model | × | 95.5±0.2 | - | 90.3±0.4 | - | 99.5±0.0 | - | 95.3±0.8 | - |

## 4 Experiments

**Datasets.** Following the GCL performance benchmark [39], four large public graph datasets are employed, including CoraFull [19], Arxiv [9], Reddit [6] and Products [9]. Specifically, CoraFull and Arxiv are citation networks, Reddit is constructed from Reddit posts, and Products is a co-purchasing network from Amazon. For all datasets, each task is set to contain only two classes [39]. Besides, for each class, the proportions of training, validation, and testing are set to be 0.6, 0.2, and 0.2 respectively.

**Competing Methods.** Two categories of state-of-the-art (SOTA) continual learning methods are employed for comparison: (1) general CIL methods: EWC [12], LwF [13], GEM [18] and MAS [2]; (2) graph CIL methods: ERGNN [44], TWP [15], SSM [41], SEM [42], CaT [17] and DeLoMe [20]. There are limited methods on task ID prediction for CIL, *e.g.*, [10,14], but they are not suited for graph data. To compare with this type of methods, we adapt the OOD detection-based CIL methods [10, 14] for GCIL (named OODCIL) (Details are in Appendix D.2). In addition, we include two baseline methods: **Fine-Tune** and **Joint**. The Fine-Tune method is a baseline that simply fine-tunes the learned model from previous tasks without continual learning techniques, while the Joint method is an oracle model that can see all graphs at all times and performs GCL on the full graphs of all tasks. We also report the results of an enhanced **Oracle Model** that is an enhanced version of the Joint method with access to the task ID of every test sample during inference.

**Implementation Details.** The proposed method is implemented under the GCL library [39]. TPP adopts a two-layer SGC [35] as the GNN backbone model with the same hyper-parameters as [42]. For task prototype construction, the number of steps $s$ in Laplacian smoothing is set to 3 by default. The number of tokens in each graph prompt, $k$, is also set to 3 across the four datasets. For each dataset, we report the average performance with standard deviations after 5 independent runs with different random seeds.

**Evaluation Metrics.** To evaluate the performance of continual learning methods, two commonly used metrics, average accuracy (AA) and average forgetting (AF) after all tasks have been learned, are adopted in our experiments. A larger AA/AF indicates better performance. An AF value of zero indicates a perfect performance involving no knowledge forgetting (*i.e.*, forget-free). The detailed definitions of AA and AF are given in Appendix D.5.

### 4.1 Main Results

**Comparison to SOTA Methods.** The results of TPP and its competing methods under the GCIL setting are shown in Table 1. From the table, we can draw the following key observations. (1) As demonstrated by the results of Fine-Tune, directly fine-tuning the learned model from previous tasks on the current task data leads to serious performance degradation because the knowledge of previous tasks could be easily overwritten by the new tasks. (2) CIL methods proposed for Euclidean

Table 2: AA and AF results of enabling existing GCIL methods with our task ID prediction (TP).

| Methods | CoraFull | | Arxiv | | Reddit | | Products | |
|---|---|---|---|---|---|---|---|---|
| | AA/%↑ | AF/%↑ | AA/%↑ | AF/%↑ | AA/%↑ | AF/%↑ | AA/%↑ | AF/%↑ |
| TWP | 62.6±2.2 | -30.6±4.3 | 6.7±1.5 | -50.6±13.2 | 8.0±5.2 | -18.8±9.0 | 14.1±4.0 | -11.4±2.0 |
| +TP | 94.3±0.9 | -1.6±0.4 | 89.4±0.4 | 0.0±0.3 | 78.0±18.5 | -0.2±0.4 | 81.8±3.3 | -0.3±0.8 |
| DeLoMe | 81.0±0.2 | -3.3±0.3 | 50.6±0.3 | 5.1±0.4 | 97.4±0.1 | -0.1±0.1 | 67.5±0.7 | -17.3±0.3 |
| +TP | 95.4±0.1 | 2.0±0.6 | 90.4±0.3 | -1.1±0.2 | 99.4±0.0 | -0.1±0.0 | 94.8±0.1 | -2.2±0.2 |

data generally do not achieve satisfactory performance for GCIL, which verifies the fact that the unique graph properties should be taken into consideration for GCIL. (3) Replay-based methods generally achieve much better performance than the other baselines, showing the effectiveness of using an external memory buffer to overcome catastrophic forgetting. However, all of them still suffer from forgetting, in addition to the inter-task separation issue. (4) The performance of OODCIL demonstrates that despite achieving impressive AF performance, current OOD detection-based CIL methods are not effective for GCIL due to the overlook of graph properties in its OOD detector and classification model. (5) Different from the baselines that involve the forgetting problem to varying extents, the proposed method TPP is a fully forget-free GCIL approach, achieving an AF value of zero across all four datasets. TPP is also consistently the best performer in AA, outperforming the best-competing method by over 18% in AA averaged over the four datasets. This superiority is attributed to the highly accurate task ID prediction module in TPP and its effective task-specific graph prompt learning (see Sec. 4.2). (6) Our method lifts the SOTA AA performance by a large margin and even significantly outperforms the oracle baseline Joint in all cases. This is because although the Joint method can mitigate catastrophic forgetting due to its access to the data of all graphs, it is still challenged by the inter-task class separation issue since it is not given task ID during inference. TPP effectively tackles both catastrophic forgetting and inter-task class separation issues, thus achieving significantly better AA than Joint and very comparable AA to the Oracle Model.

**Enabling Existing GCIL Methods with Our Task ID Prediction Module.** Existing GCIL Methods often suffer from a severe inter-task class separation issue. Our task ID prediction is devised as a module to tackle this issue. To show its effectiveness as an individual plug-and-play module, we evaluate its performance when combined with existing GCIL methods. Our task ID prediction method does not change the training process of existing GCIL models. It is directly incorporated into them at the inference stage only, *i.e.*, our task ID predictor produces a task ID for each test sample and the existing GCIL models then perform intra-task classification in the predicted task. Without loss of generality, a parameter regularization-based method (TWP [15]) and a memory-replay method (DeLoMe [20]) are used as exemplars for this experiment. The results are shown in Table 2. We can see that both AA and AF performance of the two existing GCIL models are largely enhanced by the proposed task identification module. For relatively weak GCIL models like TWP, the improvement is much more substantial than the strong ones like DeLoMe. The reason is that being able to predict the task ID accurately enables the subsequent CIL classification within the original task space of the test graph, not the space containing all the learned classes, significantly simplifying (reducing) the classification space. Essentially, such a task ID prediction converts the GCIL task into the GTIL task, so that much better AA and AF results are expected.

## 4.2 Ablation Study

**Importance of Task ID Prediction.** In GCIL, the test samples are required to be classified into one of all the learned classes. To evaluate the importance of task ID prediction that helps confine the classification space of the test samples to the classes of the predicted task, we conduct the experiments of TPP without the proposed task profiling approach and report the results in Table 3. Specifically, we obtain the class probabilities of the test sample for all tasks and prompts, and the class with the highest probability is treated as the class for the test sample. As shown in the table, this TPP variant can barely work on all four datasets. This is mainly due to that the graph prompts are learned task by task during training. Without the guidance of task identification, the non-normalized within-task prediction probabilities obtained with corresponding prompts pose great challenges for classifying the test samples into the correct classes.

**Importance of Graph Prompting.** Besides the task ID prediction, we also evaluate the importance of graph prompting. There are two modules for each task in the proposed graph promoting, *i.e.*, graph prompt $\Phi^t$ and classification head $\varphi^t$. The results with and without each module are shown in

Table 3: Results of TPP and its variants on ablating task ID prediction and graph prompting modules.

| Task ID Prediction | Graph Prompting | | CoraFull | | Arxiv | | Reddit | | Products | |
|---|---|---|---|---|---|---|---|---|---|---|
| | Prompt | Classification Head | AA/%↑ | AF/%↑ | AA/%↑ | AF/%↑ | AA/%↑ | AF/%↑ | AA/%↑ | AF/%↑ |
| ✗ | ✓ | ✓ | 2.0 | -5.5 | 3.0 | -10.9 | 2.8 | -16.8 | 2.7 | -8.2 |
| ✓ | ✗ | ✗ | 50.7 | 0.0 | 54.0 | 0.0 | 47.4 | 0.0 | 51.8 | 0.0 |
| ✓ | ✗ | ✓ | 73.8 | 0.0 | 76.3 | 0.0 | 98.6 | 0.0 | 90.0 | 0.0 |
| ✓ | ✓ | ✗ | 92.8 | 0.0 | 82.9 | 0.0 | 99.0 | 0.0 | 90.7 | 0.0 |
| ✓ | ✓ | ✓ | 93.4 | 0.0 | 85.4 | 0.0 | 99.5 | 0.0 | 94.0 | 0.0 |

Table 3. We can see that the TPP variant without both $\Phi^t$ and $\varphi^t$, which is equivalent to the direct use of the GNN backbone learned only from the first task for all subsequent tasks, achieves the worst AA performance, though it is free of forgetting since there is no model updating. By incorporating either $\Phi^t$ or $\varphi^t$, the performance can be largely improved, which can be attributed to the transferable knowledge in the pre-trained GNN $f$ and the effective adaptation of the prompts or the classifier to the subsequent tasks. Note that the variant with only $\Phi^t$ obtains much better performance than that with only $\varphi^t$, demonstrating that the learned graph prompts can more effectively model the task-specific information and bridge the gap between the first task and subsequent tasks. The results also explain the visualization of node embeddings with and without the graph prompt in Fig. 1, where the graph prompt can largely enhance the intra-task separation. By integrating all the components, the full TPP model achieves the best performance across all datasets.

**Sensitivity w.r.t the Size of Graph Prompts.** We evaluate the sensitivity of the proposed method w.r.t the size of the graph prompt, *i.e.*, the number of tokens per prompt. We vary $k$ in the range of $[1, 6]$ to verify the sensitivity and report the results in Fig. 4(a). It is clear that the performance of TPP increases quickly from $k = 1$ to $k = 2$ and remains stable when $k > 2$, demonstrating that TPP can be effectively adapted to different tasks with a small size of prompt for each task. This also demonstrates the transferability of the learned GNN backbone for all tasks.

**Accuracy of Task ID Prediction.** We further evaluate the accuracy of the proposed task ID prediction method. We compare it to a variant of our method that constructs the task prototype based on the node attributes without considering the graph structure. In this variant, each task prototype is constructed by simply averaging the attributes of training nodes of each task. The task prototype of a test graph is constructed with the test nodes in the same way. The inference process remains the same as the

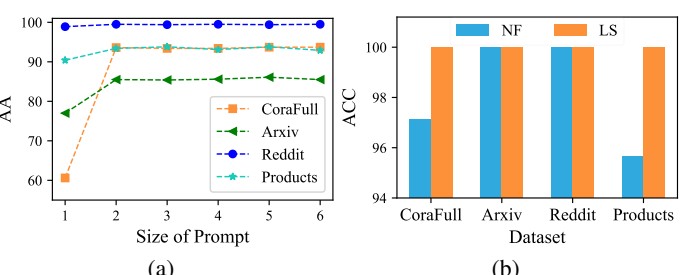

(a)                (b)

Figure 4: **(a)** The AA results of TPP w.r.t. the size of the graph prompts. **(b)** Task ID prediction accuracy on all four datasets using Laplacian smoothing (LS) and its variant based on solely node features (NF).

proposed method. The results of these two methods are shown in Fig. 4(b). We see that the task identification sorely based on node attributes achieves high accuracy for all datasets and even predicts all of the tasks correctly for Arxiv and Reddit. This is largely attributed to the discriminative node attributes between tasks in these datasets, as demonstrated in Fig. 3. However, it fails to discriminate tasks with similar node attributes. By contrast, the proposed method based on Laplacian smoothing can handle all the cases, resulting in perfect task ID prediction for all tasks and datasets, which builds a strong foundation for superior GCIL performance of TPP.

**Performance of TPP with different task formulations.** For the task formulation, we set each task to contain two different classes of nodes and follow the commonly used task formulation strategy in [20, 39] to have fair comparisons with the baselines. Specifically, given a graph dataset with several classes, we split these classes into different tasks in numerically ascending order of the original classes, i.e., classes 0 and 1 form the first task, classes 2 and 3 form the second task, and so on. To evaluate the performance of TPP with different task formulations, we further perform the class splitting in two other manners, including numerically descending and random ordering of the two classes per task. In Table 4, we report the average performance of the TPP and the Oracle Model with different task formulations.

Table 4: Results of average performance of TPP and Oracle Model on datasets with various task formulations.

| Task Formulation | Method | CoraFull | Arxiv | Reddit | Prodcuts |
|---|---|---|---|---|---|
| Ascending Order | TPP | 93.4 | 85.4 | 99.5 | 94.0 |
| Ascending Order | Oracle Model | 95.5 | 90.3 | 99.5 | 95.3 |
| Descending Order | TPP | 94.5 | 85.9 | 99.4 | 93.9 |
| Descending Order | Oracle Model | 96.1 | 91.6 | 99.5 | 94.7 |
| Random Order | TPP | 94.8 | 86.9 | 99.5 | 85.9 |
| Random Order | Oracle Model | 95.3 | 91.3 | 99.7 | 86.8 |

From the table, we observe that the proposed TPP method can still achieve comparable performance to the Oracle Model with different task formulations, highlighting the robustness and effectiveness of TPP w.r.t. the formulation of individual tasks. Note that the performances of TPP and the Oracle Model both drop on Products with random task formulation. This is attributed to the heavily imbalanced class distribution of Products and the performance is evaluated by the balanced classification accuracy. Specifically, for Products, some classes contain hundreds of thousands of nodes while the number of nodes in some classes is less than 100. The ascending and descending task formulations have a relatively balanced class distribution for each task. However, the random task formulation results in some tasks with heavily imbalanced class distribution. To address this problem, debiased learning is required and we leave it for future research. Please also note that TPP learns the GNN backbone only on the first task and is frozen during the subsequent prompt learning. Different task formulations result in the GNN backbone being learned with different first tasks. The above results also reveal that the proposed graph prompting enables the learned GNN backbone to effectively adapt to all subsequent tasks despite the backbone being learned on different initial tasks.

## 5 Conclusion

This paper proposes a novel approach for GCIL via task profiling and prompting. The absence of task IDs during inference poses significant challenges for GCIL. To address this issue, this paper proposes a novel Laplacian smoothing-based graph task profiling approach for GCIL, where each task is modeled by a task prototype based on Laplacian smoothing over the graph. We prove theoretically that the task prototypes of the same graph task are nearly the same with a large smoothing step and the prototypes of different tasks are distinct due to the differences in graph structure and node attributes, ensuring accurate task ID prediction for GCIL. To avoid catastrophic forgetting and achieve high within-task prediction, we further propose the first graph prompting method for GCIL which is learned to absorb the within-task information into the small task-specific graph prompts. This results in a memory-efficient TPP as i) no memory buffer is required for data replay due to its replay-free characteristic and ii) the graph prompting only requires the training of a single GNN once and a small number of tokens per prompt for each task. Extensive experiments show that TPP is fully forget-free and significantly outperforms the state-of-the-art baselines for GCIL.

## Acknowledgments

In this work, the participation of Chaoxi Niu and Ling Chen was supported by Australian Research Council under Grant DP210101347, while the participation of Guansong Pang was supported in part by Lee Kong Chian Fellowship. The work of Bing Liu was supported in part by four NSF grants (IIS-2229876, IIS-1910424, IIS-1838770, and CNS-2225427).

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

# A Proof of Theorems

**Theorem 1.** *If graphs for all tasks are not isolated and the test graph $\mathcal{G}^{test}$ comes from the task t, i.e., $\mathcal{G}^{test}$ and $\mathcal{G}^t$ have the same set of classes, then the distance between $\mathbf{p}^{test}$ and $\mathbf{p}^t$ approaches to zero with a sufficiently large number of Laplacian smoothing steps s:*

$$\lim_{s \to +\infty} d(\mathbf{p}^{test}, \mathbf{p}^t) = 0. \tag{6}$$

*Proof.* To prove the distance between $\mathbf{p}^{\text{test}}$ and $\mathbf{p}^t$ approaches 0 with a large Laplacian smoothing step $s$, we need to illustrate that the features of nodes in $\mathcal{G}^t$ coverage to be proportional to the square root of the node degree after Laplacian smoothing and the proposed task prototype construction method transforms all node features to the same values. Note that the self-loop is added to each node in the graph, resulting in the graph being non-bipartile. Assuming the size of the graph $\mathcal{G}^t$ is $N$, the Laplacian matrix $(\hat{D}^t)^{-\frac{1}{2}} \hat{L}^t (\hat{D}^t)^{-\frac{1}{2}}$ has $N$ eigenvalues with different eigenvectors [29]. Recalling the Laplacian smoothing defined in Eq. (2), the eigenvalues and eigenvectors of $I - (\hat{D}^t)^{-\frac{1}{2}} \hat{L}^t (\hat{D}^t)^{-\frac{1}{2}}$ can be represented as $(\lambda_1, \ldots, \lambda_N)$ and $(\mathbf{e}_1, \ldots, \mathbf{e}_N)$ respectively. With the property of symmetric Laplacian matrix for the non-bipartile graph, the eigenvalues of $I - (\hat{D}^t)^{-\frac{1}{2}} \hat{L}^t (\hat{D}^t)^{-\frac{1}{2}}$ are all in the range of $(-1, 1]$ [3], *i.e.,*

$$-1 < \lambda_1 < \ldots < \lambda_N = 1, \quad \mathbf{e}_N = (\hat{D}^t)^{\frac{1}{2}} [1, 1, \ldots, 1]^T \in \mathbb{R}^{N \times 1}. \tag{13}$$

Based on the eigenvalues and the eigenvectors, the result of applying Laplacian smoothing on the node features $X^t$ after $s$ steps can be formulated as:

$$(I - (\hat{D}^t)^{-\frac{1}{2}} \hat{L}^t (\hat{D}^t)^{-\frac{1}{2}})^s X^t = [\lambda_1^s \mathbf{e}_1, \ldots, \lambda_N^s \mathbf{e}_N] \hat{X}^t, \tag{14}$$

where $\hat{X}^t = [\mathbf{e}_1, \ldots, \mathbf{e}_N]^{-1} X^t$. As the eigenvectors are orthogonal to each other, we can further rewrite $\hat{X}^t$ as $\hat{X}^t = [\mathbf{e}_1, \ldots, \mathbf{e}_N]^T X^t$. Since the absolute values of the eigenvalues are less than 1 except $\lambda_N$, $\lambda_i^s$ would approach 0 as $s$ go infinity, *i.e.,* $\lim_{s \to \infty} \lambda_i^s = 0, \forall i \neq N$. Then, we can formulate the smoothed node feature representations $Z^t$ as follows:

$$Z^t = [(\hat{D}_{11}^t)^{\frac{1}{2}}, \ldots, (\hat{D}_{NN}^t)^{\frac{1}{2}}]^T \hat{X}^t[N, :], \tag{15}$$

where $\hat{X}^t[N, :]$ denotes the $N$-th row of $\hat{X}^t$. Therefore, with a larger $s$, the feature of nodes in $\mathcal{G}^t$ would converge to be proportional to the square root of the node degree. By multiplying the smoothed feature with $(\hat{D}_{ii}^t)^{-\frac{1}{2}}$ for each node $i$ in the proposed task prototype construction, $\mathbf{p}^{\text{test}}$ and $\mathbf{p}^t$ would both become to be $\hat{X}^t[N, :]$, despite that they utilize different nodes to construct the task prototypes. Therefore, the distance between $\mathbf{p}^{\text{test}}$ and $\mathbf{p}^t$ would become zero with a large $s$. $\square$

Note that we assume that the graphs of all tasks are not isolated in Theorem 1. Having isolated nodes in real-world graphs would deviate from this assumption, resulting in unsatisfied task identification. To tackle this issue, we propose a simple graph augmentation when constructing the task prototypes, which adds an edge between the isolated nodes and the randomly chosen non-isolated nodes to make the graph more connected. This helps the proposed task ID prediction method predict the task of test graphs more accurately.

**Theorem 2.** *Suppose the test graph comes from task t, and let $\mathbf{e}$ and $\epsilon$ be the differences in node degrees and node attributes between two different tasks t and j respectively, which are defined by $(\hat{D}^j)^{\frac{1}{2}} = (\hat{D}^t)^{\frac{1}{2}} + \text{Diag}(\mathbf{e})$ and $X^j = X^t + \epsilon$. Then the distance between the task prototypes of task t and j obtained with large steps of Laplacian smoothing can be explicitly calculated as:*

$$d(\mathbf{p}^{test}, \mathbf{p}^j) = \|(\mathbf{e}_N^t)^T \epsilon + (\mathbf{e})^T X^j\|_2, \tag{7}$$

*where $\mathbf{e}_N^t = (\hat{D}^t)^{\frac{1}{2}} [1, 1, \ldots, 1]^T$ is the N-th eigenvector of task t and $(\mathbf{e}_N^t)^T$ denotes its transpose.*

*Proof.* As derived in Theorem 1, the task prototypes of task $t$ and $j$ with large steps of Laplacian smoothing are $\mathbf{p}^{\text{test}} = \hat{X}^t[N, :]$ and $\mathbf{p}^j = \hat{X}^j[N, :]$ respectively. Furthermore, the task prototype

of task $t$ can be represented as $\mathbf{p}^t = (\mathbf{e}_N^t)^T X^t$. Based on the difference in node degrees and node attributes between task $t$ and $j$, the distance between the task prototypes can be represented as follows:

$$d(\mathbf{p}^{\text{test}}, \mathbf{p}^j) = \|\mathbf{p}^t - \mathbf{p}^j\|_2 \tag{16}$$

$$= \|(\mathbf{e}_N^t)^T X^t - (\mathbf{e}_N^j)^T X^j\|_2 \tag{17}$$

$$= \|(\mathbf{e}_N^t)^T X^t - (\mathbf{e}_N^t + \mathbf{e})^T (X^t + \epsilon)\|_2 \tag{18}$$

$$= \|(\mathbf{e}_N^t)^T \epsilon + (\mathbf{e})^T X^j\|_2 \tag{19}$$

$$\square$$

## B  Details on Learning GNN Backbone

We propose to construct a GNN backbone $f(\cdot)$ for graph prompt learning so that the task-specific information can be absorbed into the prompts. Specifically, the model $f(\cdot)$ is constructed based on the first task $\mathcal{G}^1 = (A^1, X^1)$ via graph contrastive learning due to its ability to obtain transferable models [36, 45] across graphs.

To construct contrastive views for graph contrastive learning, two widely used graph augmentations are employed, *i.e.*, edge removal and attribute masking [45]. Specifically, the edge removal randomly drops a certain portion of existing edges in $\mathcal{G}^1$ and the attribute masking randomly masks a fraction of dimensions with zeros in node attributes, *i.e.*,

$$\tilde{A}^1 = A^1 \circ R, \quad \tilde{X}^1 = [\mathbf{x}_1^1 \circ \mathbf{m}, \ldots, \mathbf{x}_N^1 \circ \mathbf{m}]^T, \tag{20}$$

where $R \in \{0,1\}^{N \times N}$ is the edge masking matrix whose entry is drawn from a Bernoulli distribution controlled by the edge removal probability, $\mathbf{m} \in \{0,1\}^F$ is the attribute masking vector whose entry is independently drawn from a Bernoulli distribution with the attribute masking ratio, and $\circ$ denotes the Hadamard product. By applying the graph augmentations to the original graph, the corrupted graph $\tilde{\mathcal{G}}^1 = (\tilde{A}^1, \tilde{X}^1)$ forms the contrastive view for the original graph $\mathcal{G}^1 = (A^1, X^1)$. Then, $\tilde{\mathcal{G}}^1$ and $\mathcal{G}^1$ are inputted to the shared GNN $f(\cdot)$ followed by non-linear projection $g(\cdot)$ to obtain the corresponding node embeddings, *i.e.*, $\tilde{Z}^1 = g(f(\tilde{\mathcal{G}}^1))$ and $Z^1 = g(f(\mathcal{G}^1))$. For graph contrastive learning, the embeddings of the same node in different views are pulled closer while the embeddings of other nodes are pushed apart. The pairwise objective for each node pair $(\tilde{\mathbf{z}}_i^1, \mathbf{z}_i^1)$ can be formulated as:

$$\ell(\tilde{\mathbf{z}}_i^1, \mathbf{z}_i^1) = -\log \frac{e^{sim(\tilde{\mathbf{z}}_i^1, \mathbf{z}_i^1)/\tau}}{e^{sim(\tilde{\mathbf{z}}_i^1, \mathbf{z}_i^1)/\tau} + \sum_{j \neq i}^N e^{sim(\tilde{\mathbf{z}}_i^1, \mathbf{z}_j^1)/\tau} + \sum_{j \neq i}^N e^{sim(\tilde{\mathbf{z}}_i^1, \tilde{\mathbf{z}}_j^1)/\tau}}, \tag{21}$$

where $sim(\cdot)$ represents the cosine similarity and $\tau$ is a temperature hyperparameter. Therefore, the overall objective can be defined as follows:

$$\mathcal{L}_{\text{contrast}} = \frac{1}{2N} \sum_{i=1}^N (\ell(\tilde{\mathbf{z}}_i^1, \mathbf{z}_i^1) + \ell(\mathbf{z}_i^1, \tilde{\mathbf{z}}_i^1)). \tag{22}$$

With the objective Eq. (22), the model $f(\cdot)$ is optimized to learn discriminative representations of nodes. Despite the limited size of the first task, the learned model $f(\cdot)$ can effectively adapt to other tasks with the proposed graph prompt learning method.

## C  Algorithm

The training and inference processes of the proposed method are summarized in Algorithm 1 and Algorithm 2, respectively.

## D  Experimental Setup

### D.1  More Implementation Details

All the continual learning methods including the proposed method are implemented based on the GCL benchmark [39][2]. For the memory-replay methods, we follow the settings in [20]. As in [42],

---

[2]https://github.com/QueuQ/CGLB/tree/master

---

**Algorithm 1:** Training of TPP

---

1: **Input:** A series of graph learning tasks: $\{\mathcal{G}^1, \ldots, \mathcal{G}^T\}$, a graph neural network $f(\cdot)$.
2: **Output:** Graph neural network $f(\cdot)$, task prototype $\mathcal{P} = \{\mathbf{p}^1, \ldots, \mathbf{p}^T\}$, graph prompts $\{\Phi^1, \ldots, \Phi^T\}$, and classifiers $\{\varphi^1, \ldots, \varphi^T\}$.
3: Pre-train $f(\cdot)$ on $\mathcal{G}^1$ with graph contrastive learning (Eq. (22)).
4: **for** $t = 1, \ldots, T$ **do**
5:     Obtain the task prototype $\mathbf{p}^t$ of task $t$ (Eq. (3)).
6:     Obtain $\bar{\mathcal{G}}^t$ with graph prompts $\Phi^t$ (Eq. (9)).
7:     Optimize $\Phi^t$ and $\varphi^t$ by minimizing node classification loss (Eq. (11)).
8: **end for**

---

---

**Algorithm 2:** Inference in TPP

---

1: **Input:** Graph neural network $f(\cdot)$, task prototypes $\mathcal{P} = \{\mathbf{p}^1, \ldots, \mathbf{p}^T\}$, graph prompts $\{\Phi^1, \ldots, \Phi^T\}$, classifiers $\{\varphi^1, \ldots, \varphi^T\}$, and the test graph $\mathcal{G}^{\text{test}}$.
2: **Output:** Prediction result.
3: Obtain the task prototype $\mathbf{p}^{\text{test}}$ (Eq. (4)).
4: Infer the task $t^{\text{test}}$ of the test graph by querying $\mathcal{P}$ with $\mathbf{p}^{\text{test}}$ (Eq. (5)).
5: Retrieve the corresponding graph prompt $\Phi^{t^{\text{test}}}$ and classifier $\varphi^{t^{\text{test}}}$
6: Obtain $\bar{\mathcal{G}}^{\text{test}} = (A^{\text{test}}, X^{\text{test}} + \Phi^{t^{\text{test}}})$ (Eq. (9)).
7: **return** $\varphi^{t^{\text{test}}}(f(A^{\text{test}}, X^{\text{test}} + \Phi^{t^{\text{test}}}))$

---

we employ a two-layer SGC [35] model as the backbone. Specifically, the hidden dimension is set to 256 for all methods. The number of training epochs of each graph learning task is 200 with Adam as the optimizer and the learning rate is set to 0.005 by default.

For graph contrastive learning, the probabilities of edge removal and attribute masking are set to 0.2 and 0.3 respectively for all datasets. Besides, the learning rate is set to 0.001 with Adam optimizer, the training epochs are set to 200 and the temperature $\tau$ is 0.5 for all datasets.

The code is implemented with Pytorch (version: 1.10.0), DGL (version: 0.9.1), OGB (version: 1.3.6), and Python 3.8.5. Besides, all experiments are conducted on a Linux server with an Intel CPU (Intel Xeon E-2288G 3.7GHz) and a Nvidia GPU (Quadro RTX 6000).

### D.2 Details on the Design of the OODCIL Method

We adapt the OOD detection-based CIL methods in [10,14] for empirical comparison under GCIL. To this end, following [10,14], we propose to build an OOD detector for each graph task to perform task ID prediction and within-task classification simultaneously. Specifically, for task $t$ with $C$ classes, we aim to learn an OOD detector $f_o^t(\cdot)$ with $C+1$ classes. The extra class represents the OOD data for this task. In this paper, we implement the OOD detector as a two-layer SGC [35] model and take the data in replay buffer $Buf_{<t}$ as the OOD data for task $t$. Formally, the OOD detector is optimized by minimizing the following objective for task $t$:

$$\min_{f_o^t(\cdot)} \mathbb{E}_{\mathcal{G}^t \bigcup Buf_{<t}} \left[ \ell_{\text{CE}}(f_o^t(\mathcal{G}^t), Y^t) + \ell_{\text{CE}}(f_o^t(Buf_{<t}), Y^{Buf}) \right] , \tag{23}$$

where $Y^{Buf} = C+1$ represents the labels of data in the replay buffer $Buf_{<t}$. The buffer $Buf_{<t}$ is constructed via sampling previous graphs following ERGNN [44]. Note that there are no replay data for task 1 to train the OOD detector. To overcome this issue, we propose to generate OOD data for task 1 based on graph augmentation [45].

After learning all the tasks, we follow Eq. (1) to compute the class probabilities for the test samples. Specifically, a test sample is fed into all the learned OOD detectors to obtain the OOD score and class probabilities within the corresponding tasks. For example, for task $t$, the learned OOD detector would output the class and OOD probabilities $\{y_1^t, \ldots, y_C^t, o^t\}$ for the test sample. As the OOD score indicates the probability of the test sample is OOD to this task, the final probabilities of the test sample w.r.t. the classes in task $t$ can be calculated as $\{(1-o^t)y_1^t, \ldots, (1-o^t)y_C^t\}$. Among all the class probabilities of all tasks, the class with the highest probability is predicted as the class for the test sample.

Table 5: Key statistics of the graph datasets.

| Datasets | CoraFull | Arxiv | Reddit | Products |
|---|---|---|---|---|
| # nodes | 19,793 | 169,343 | 227,853 | 2,449,028 |
| # edges | 130,622 | 1,166,243 | 114,615,892 | 61,859,036 |
| # classes | 70 | 40 | 40 | 46 |
| # tasks | 35 | 20 | 20 | 23 |
| # Avg. nodes per task | 660 | 8,467 | 11,393 | 122,451 |
| # Avg. edges per task | 4,354 | 58,312 | 5,730,794 | 2,689,523 |

## D.3 Details on Datasets

Following [39], four large GCIL datasets are used in our experiments.

- **CoraFull**[3]: It is a citation network containing 70 classes, where nodes represent papers and edges represent citation links between papers.
- **Arxiv**[4]: It is also a citation network between all Computer Science (CS) ARXIV papers indexed by MAG [23]. Each node in Arxiv denotes a CS paper and the edge between nodes represents a citation between them. The nodes are classified into 40 subject areas. The node features are computed as the average word-embedding of all words in the title and abstract.
- **Reddit**[5]: It encompasses Reddit posts generated in September 2014, with each post classified into distinct communities or subreddits. Specifically, nodes represent individual posts, and the edges between posts exist if a user has commented on both posts. Node features are derived from various attributes, including post title, content, comments, post score, and the number of comments.
- **Products**[6]: It is an Amazon product co-purchasing network, where nodes represent products sold in Amazon and the edges between nodes indicate that the products are purchased together. The node features are constructed with the dimensionality-reduced bag-of-words of the product descriptions.

The statistics of the used graph datasets are summarized in Table 5.

## D.4 Descriptions of Baselines

- **EWC** [12] is a regularization-based method that adds a quadratic penalty on the model parameters according to their importance to the previous tasks to maintain the performance on previous tasks.
- **MAS** [2] preserves the parameters important to previous tasks based on the sensitivity of the predictions to the changes in the parameters.
- **GEM** [18] stores representative data in the episodic memory and proposes to modify the gradients of the current task with the gradient calculated on the memory data to tackle the forgetting problem.
- **LwF** [13] employs knowledge distillation to minimize the discrepancy between the logits of the old and the new models to preserve the knowledge of the previous tasks.
- **TWP** [15] proposes to preserve the important parameters in the topological aggregation and loss minimization for previous tasks via regularization terms.
- **ERGNN** [44] is a replay-based method that constructs memory data by storing representative nodes selected from previous tasks.
- **SSM** [41] incorporates the explicit topological information of selected nodes in the form of sparsified computation subgraphs into the memory for graph continual learning.
- **SEM** [42] improves SSM by storing the most informative topological information via the Ricci curvature-based graph sparsification technique.

---

[3] https://docs.dgl.ai/en/1.1.x/generated/dgl.data.CoraFullDataset.html
[4] https://ogb.stanford.edu/docs/nodeprop/#ogbn-arxiv
[5] https://docs.dgl.ai/en/1.1.x/generated/dgl.data.RedditDataset.html#dgl.data.RedditDataset
[6] https://ogb.stanford.edu/docs/nodeprop/#ogbn-products

Table 6: The accuracy of task prediction with other task formulations.

| Task Formulation | CoraFull | Arxiv | Reddit | Prodcuts |
|---|---|---|---|---|
| Ascending Order (Reported) | 100 | 100 | 100 | 100 |
| Descending Order | 100 | 100 | 100 | 100 |
| Random Order | 100 | 100 | 100 | 100 |

Table 7: Additional parameters and performance (AA%) of the proposed graph prompting and task-specific models.

| Method | Additional Parameters | CoraFull | Arxiv | Reddit | Products |
|---|---|---|---|---|---|
| Task-specific Models | $(F + d)dT$ | 94.3 | 86.8 | 99.5 | 96.3 |
| Graph Prompting | $3FT$ | 93.4 | 85.4 | 99.5 | 94.0 |

- **CaT** [17] condenses each graph to a small synthesized replayed graph and stores it in a condensed graph memory with historical replay graphs. Moreover, graph continual learning is accomplished by updating the model directly with the condensed graph memory.

- **DeLoMe** [20] learns lossless prototypical node representations as the memory to capture the holistic graph information of previous tasks. A debiased GCL loss function is further devised to address the data imbalance between the classes in the memory data and the current data.

### D.5 Evaluation Metrics: Average Accuracy and Average Forgetting

Specifically, average accuracy (AA) and average forgetting (AF) are calculated from the lower diagonal accuracy matrix $M \in \mathbb{R}^{T \times T}$, where $T$ is the number of the tasks. The entry $M_{tj}(t \geqslant j)$ denotes the classification accuracy on task $j$ after the model is optimized on task $t$. Therefore, the row in $M_{t,:}$ records the performance on all previous tasks after learning task $t$, and the column $M_{:,j}$ describes the dynamic of performance on task $j$ when learning different tasks. After learning all the $T$ tasks, the overall average accuracy (AA) and average forgetting (AF) can be calculated as follows:

$$\text{AA} = \frac{\sum_{j=1}^{T} M_{Tj}}{T}, \quad \text{AF} = \frac{\sum_{j=1}^{T-1}(M_{Tj} - M_{jj})}{T - 1}. \tag{24}$$

To sum up, AA evaluates the average performance of the model on all the learned tasks after learning all the $T$ tasks, and AF describes how the performance of previous tasks is affected when learning the current task. A positive AF indicates learning the current task would facilitate the previous tasks and vice versa. For both AA and AF, the higher value denotes better GCL performance.

### D.6 More Experimental Results

**Accuracy of task prediction with different task formulations.** We further evaluate the accuracy of task prediction with different task formulations, *i.e.*, numerically descending and random ordering. The results are shown in Table 6. The results demonstrate that the proposed TP can accurately predict the task IDs in terms of all formulations.

**Comparison to Task-Specific Models** As discussed in the main paper, another straightforward way to overcome forgetting is to learn task-specific models for each task. We further compare the number of additional parameters and the performance of the proposed graph prompting with task-specific models besides the parameters of the backbone. The two methods can both achieve forget-free for GCIL with the proposed task identification. The results are reported in Table 7 where $F$ is the dimensionality of the node attribute, $d$ is the number of hidden units of SGC and $T$ denotes the number of tasks. From the table, we can see that the proposed methods can achieve very close performance to task-specific models while introducing significantly small additional parameters for all tasks in GCIL.

## E  Time Complexity Analysis

The proposed method first learns a GNN backbone based on the first task with graph contrastive learning. Then, the model remains frozen and the task-specific graph prompts and classifiers are

Table 8: Total training time and inference time (seconds) for different methods on CoraFull.

| Methods | TWP | SSM | DeMoLe | TPP (Ours) |
|---|---|---|---|---|
| Training Time | 151.6 | 254.9 | 304.6 | 23.6 |
| Inference Time | 0.3 | 0.3 | 0.3 | 0.4 |

optimized to capture the knowledge of each task separately. In experiments, we employ a two-layer SGC [35] as the GNN backbone model with the number of hidden units in all layers as $d$. Suppose all tasks contain the same number of nodes as $N$, the time complexity of the graph contrastive learning on the first task is $\mathcal{O}((4|A^1|F + 2NdF + 3Nd^2)E_1)$, where $|A^1|$ returns of the number of edges of the $\mathcal{G}^1$, $F$ represents the dimensionality of node attributes and $E_1$ is the number of training epochs. After that, we propose to freeze the learned model and learn graph prompts and classifiers for each task. In our experiments, we set the size of each graph prompt to $k$ and implement the classification head as a single-layer MLP outputting the probabilities of $C$ classes. Given the number of the training epochs $E_2$, the time complexity of optimizing the graph prompt and classifier is $\mathcal{O}((4kNF + 2dNC)E_2)$, which includes both the forward and backward propagation. Despite the graph model being frozen, the forward and backward propagation of the model are still needed to optimize the task-specific graph prompts and classifiers. Given the number of tasks $T$ in GCIL, the overall time complexity of the proposed method is $\mathcal{O}((4|A^1|F + 2NdF + 3Nd^2)E_1 + \sum_{i=1}^{T}(4|A^1|F + 2NdF + 3Nd^2 + 4kNF + 2dNC)E_2)$, which is linear to the number of nodes, the number of edges, and the number of node attributes involved in all the graph tasks.

In Table 8, we report the total training time and inference time of all tasks on the CoraFull dataset, with representative models TWP [15], SSM [41] and DeLoMe [20] as the baselines, where TWP is a regularization-based method and the other two baselines are replay-based methods. From the table, we can see that replay-based methods require more time for training. This is because they typically need to construct the memory buffer based on different strategies for replaying with the new graph data. Moreover, these memory buffers accumulate to store information from all learned tasks, resulting in the size of them becoming larger with more tasks learned. Notably, our method requires the smallest amount of training time as it does not introduce replaying memory and regularization terms. As for the inference time, the three baselines require the same amount of time as they all learn a model for all tasks. Our method requires slightly more inference time due to its task ID prediction module. Thus, there is a computational overhead in the inference of our method TPP, but it is trivial.

## F Limitations

This paper investigates graph class-incremental learning with task identification and graph prompting. The task identification is achieved by modeling each graph task with task prototypes based on Laplacian smoothing. We theoretically and empirically demonstrate that task identification can be accurately performed across graphs. This helps address the inter-task class separation issue. To overcome the catastrophic forgetting problem, we propose a graph-prompting approach that absorbs within-task discriminative information into small task-specific graph prompts. The proposed method achieves significant performance improvement. One key limitation lies in the limited representative capacity and generalizability of the GNN backbone model in prompting. This paper constructs the GNN based on the first task for each dataset, resulting in the capacity of the model being constrained by the size of the first task and not applicable to other datasets directly, *i.e.*, a different GNN backbone needs to be trained separately for each dataset. In our future work, we aim to explore approaches to learn more strong GNN backbones that are transferable across different datasets.

## G Broader Impacts

Graph continual learning aims to continually learn a model that not only accommodates the new emerging graph data but also maintains the learned knowledge of previous tasks. It eliminates the need to retrain the model on all data when new data emerges, significantly reducing computational costs in real-world applications. This paper studies the challenging graph class-incremental learning and proposes a novel, memory-efficient, and forget-free method that is easy to learn and computationally efficient (and thus eco-friendly). Additionally, the proposed method does not require storing previous data for replay, thereby preserving data privacy.

