# OpenReview forum: "Replay-and-Forget-Free Graph Class-Incremental Learning: A Task Profiling and Prompting Approach"
_NeurIPS.cc/2024/Conference — NeurIPS 2024 poster_

### Official Review · Reviewer_aHCQ · 2024-06-26

**Soundness:** 3
**Presentation:** 3
**Contribution:** 3
**Rating:** 7
**Confidence:** 5

**Summary:**

- The paper tackles the problem of graph class-incremental learning. The proposed TPP consists of two modules, that is, task profiling and graph prompting.
   - The task prediction is accomplished by learning task prototypes based on graph Laplacian smoothing. Specifically, task profiling aims to accurately predict the task ID of each task during inference by constructing Laplacian smoothing-based task prototypes.
   - Graph prompting aims to capture task-specific knowledge into graph prompts. Graph prompting is to avoid the catastrophic forgetting of the knowledge learned in previous graph tasks, which learns a small discriminative graph prompt for each task.
- TPP shows significant performance improvements, as demonstrated by experiments on four graph datasets.

**Strengths:**

- The idea of using Laplacian smoothing to predict the task IDs and graph prompts to capture task-specific knowledge is interesting and new in graph class-incremental learning.
- The proposed method, TPP, is novel, which not only predicts the task IDs of test graph tasks but also distills task-specific information into prompts.
- The proposed method is both replay-free and forget-free and requires the training of a single GNN only once.
- The task IDs of each graph task can be accurately predicted with theoretical support.
- The paper is clearly motivated and well-written.

**Weaknesses:**

- The task prediction relies on the graph transduction setting. The paper focuses on the subgraph incremental learning without considering the inter-edges between graph tasks, resulting in the proposed method not generalizable to other settings effectively.
- Not clear definitions of different categories of methods for graph continual learning. The authors argue that graph prompting can reduce the heavy burdens on optimization and storage with the increasing number of tasks compared to training a separate GNN for each task. However, there are no empirical comparisons.
- Some ablation studies are missing.

**Questions:**

- The authors should point out the key differences of task prediction between image and graph more clearly.
- The authors should include the comparisons between graph prompting and training separate GNNs for each task. Can the proposed task identification method be used for other domains like continual image classification? Since the authors argue that graph prompting can reduce the heavy burdens on optimization and storage compared to training a separate GNN for each task, the authors should include the comparisons between graph prompting and training separate GNNs for each task.
- In Table 3, why is the performance of TPP without task prediction significantly low as the graph prompts learn task-specific knowledge?
- In Fig. 4(a), the performance of TPP becomes stable when the size of the graph prompts is larger than one. The authors should provide a more detailed analysis on this phenomenon. The task prediction relies on the graph transduction setting and the accuracy of task prediction with training nodes can also achieve very high accuracy as shown in Fig. 4(b). Therefore, the task identification can be simply conducted by averaging all the node features as the task prototypes. The authors should compare the proposed method with this approach.
- The authors are encouraged to include clear definitions of three categories of methods for graph continual learning to make the paper more comprehensive. Besides, the descriptions of the baselines should be also included.

**Limitations:**

Yes, they addresed.

---

> ### Author Rebuttal · Authors · 2024-08-07
>
> Thank you very much for the constructive comments and questions. We are grateful for the positive comments on our design and empirical justification. Please see our detailed one-by-one responses below. We will include the new results and discussions below into the paper.
>
> >**Questions #1:** Point out the key differences of task prediction between image and graph more clearly.
>
> The major difference between images and graphs is that images are i.i.d while graphs are non-i.i.d due to the complex structure of graphs. The graph structure brings a unique challenge for the task prediction of graph continual learning. Most existing task prediction methods for images rely on an OOD detector by treating data from other tasks as the OOD data. In this paper, we utilize the graph structure to perform task prediction by using a Laplacian smoothing-based method to profile each graph task with a prototypical embedding and achieve accurate task prediction.
>
> >**Questions #2/Weaknesses #3:**  Include the comparisons between graph prompting and training separate GNNs for each task.
>
> In the following table, we report the comparison of TPP and training separate GNNs for each task in terms of additional parameters and the average accuracy, where $F$ is the dimensionality of the node attributes, $d$ is the number of hidden units in an SGC layer and $T$ denotes the number of tasks. From the table, we can see that the proposed TPP method can achieve very close performance to its variant that trains separate GNNs for each task rather than task-specific prompts, while it involves a significantly smaller number of parameters for all tasks in GCIL.
>
> ```
> Table A1. Additional parameters and the average performance of the proposed graph prompting and task-specific models.
> ```
> |Method | Additional Parameters |CoraFull | Arxiv | Reddit | Products |
> |---|---|---|---|---|---|
> |Separate Models| $(F+d)dT$|94.3| 86.8| 99.5|96.3|
> |TPP|$3FT$ |93.4|85.4|99.5|94.0|
>
> The proposed task prediction method cannot be directly used for other domains if there is an absence of graph structure information between data instances. A similar prototype-based task profiling approach can be explored for data types like images, but designs for generating more discriminative task profiles may be needed, which would be an interesting future research extension to our method TPP.
>
>
>
> >**Questions #3:** Why is the performance of TPP without task prediction significantly low as the graph prompts learn task-specific knowledge?
>
> Despite graph prompts capturing the task-specific knowledge, they are learned independently for each task and are not compatible with other tasks. Without the guidance of task identification, node classification is accomplished by concatenating the class probabilities of the test sample for all tasks and choosing the class with the highest probability as the predicted class. Specifically, assume there are $T$ tasks and each task contains $C$ classes, the predicted class of a test sample can be formulated as $c = \arg \max (c_1, c_2, \ldots, c_{T \times C})$. Suppose the test sample comes from task $i$, the incompatibility between the test sample and prompts from different tasks ($j, j\neq i$) results in the class of the test sample being biased to one class of task $j$ with an extremely high probability. As a result, the class of the test sample cannot be predicted correctly.
>
>
>
> >**Questions #4/Weaknesses #1 and #3:** Explain the performance of TPP with the size of graph prompts and discuss the task prediction using all node attributes.
>
> In our experiments, each task contains 2 classes (i.e., $C=2$) of nodes and we reported the performance of TPP with different sizes of the graph prompts in Figure 4. The figure shows that the performance of TPP increases quickly from $k = 1$ to $k = 2$ and remains stable for $k > 2$. We attribute this phenomenon to the intuition that the number of learnable tokens in graph prompts with a maximum number of $C$ is often sufficient to instruct the backbone to perform subsequent tasks conditionally. To further evaluate this intuition, we set the number of classes in each task to three ($C=3$) and report the performance of TPP with varying sizes of prompts in the following table. We can see that the performance becomes stable when $k=2$ and $k=3$ for CoraFull and Arxiv respectively, which supports our intuition.
>
> ```
> Table A2. The average results with different sizes of the graph prompts when each task consists of nodes from three classes.
> ```
> |Datasets|1|2|3|4|5|6|
> |---|---|---|---|---|---|--|
> |CoraFull|48.9|88.0|89.3|90.2|91.0|91.2|
> |Arxiv|64.3|72.2|75.9|76.1|77.0|76.7|
>
> For task prediction by averaging all the node attributes as the task prototypes, it can also achieve good task prediction for most of the datasets (please see the results with the smoothing step set to zero in our response to the Weakness \#4 of Reviewer \#3), but it is not as effective as our proposed TP method. This may be attributed to various situations where this simpler alternative approach fails to work. For example, it cannot distinguish different tasks that may have distinct node attributes but have similar averaged attribute-based prototypes. Moreover, neglecting the graph structure would also result in failures in various other cases, e.g., when different tasks share similar node attributes but have different graph structures. In contrast, the proposed method can address all these cases as shown in Eq.(7).
>
>
>
> >**Questions #5/Weaknesses #2:** Include clear definitions of three categories of methods for graph continual learning and the descriptions of baselines.
>
> Existing GCL methods can be roughly divided into three categories, i.e., regularization-based, parameter isolation-based, and data replay-based methods. We will include more detailed definitions of these categories and the descriptions of the baselines in the revision.

---

### Official Review · Reviewer_j8FV · 2024-07-11

**Soundness:** 1
**Presentation:** 3
**Contribution:** 2
**Rating:** 5
**Confidence:** 3

**Summary:**

This paper addresses the challenge of class-incremental learning (CIL) in graph data (GCIL) by proposing a novel Task Profiling and Prompting (TPP) approach. It leverages Laplacian smoothing-based task profiling to achieve accurate task ID prediction, thereby mitigating inter-task class separation issues, and introduces a graph prompting method to prevent catastrophic forgetting without the need for data replay. Extensive experiments on four benchmarks demonstrate that TPP achieves 100% task ID prediction accuracy and significantly outperforms state-of-the-art methods in average CIL accuracy while being fully forget-free.

**Strengths:**

- This paper highlights the significant challenge in graph class incremental learning (GCIL).
- It introduces a novel approach in GCIL, task ID prediction, by devising Laplacian smoothing-based task profiling for graph task ID prediction, which shows surprisingly perfect performance.
- Furthermore, extensive experiments are conducted to demonstrate the superiority of the proposed method.

**Weaknesses:**

- The proposed graph prompting method appears to merely apply the technique from [1] to the graph domain, lacking technical novelty.
- The proposed TP method raises several significant concerns:
  - The proposed TP method consists of Laplacian smoothing and average pooling without any training. Since Laplacian smoothing and GCN are significantly similar, there is no clear reason why the GCN-based OOD detector (in OODCIL) should underperform compared to the proposed TP. I kindly ask the authors to compare the performance of the GCN-based OOD detector and provide an explanation of the results, with exact numerical values rather than charts. For a similar reason, I do not understand why OODCIL significantly underperforms compared to most baselines. Therefore, please provide the following two sets of results: 1) replacing the task ID prediction module in OODCIL with TP, and 2) replacing the task ID prediction module in TPP with the OOD detector from OODCIL.
  - The zero AF scores indicate that the proposed TP perfectly classifies the task ID, which raises suspicions about the task formulation. Specifically, I suspect that each task was formulated in a way that gives TPP an advantage, making it easier to predict the task ID. Therefore, I kindly ask the authors to provide experiments under different task formulations, such as: 1) randomly sampling classes for each task, and 2) splitting the classes in numerical order by the given class number (e.g., (0,1), (2,3), (4,5),...,) and so on.

- The absence of available source code to verify the results amplifies my suspicions about the experimental results.

- In Theorems 1 and 2, the authors assume a sufficiently large number of Laplacian smoothing steps (i.e., $s \rightarrow \infty$). However, this assumption seems flawed because, as $s$ increases, the prototype will collapse due to the oversmoothing problem, making each task indiscriminative. Furthermore, the authors set the value of $s$ to 3 in their experimetns, which is far from a sufficiently large number of steps. I kindly ask the authors to provide experimental results for the accuracy of the task ID prediction module with the value of $s$ varying from {3, 5, 7, ...}. These results should demonstrate that as $s$ increases, the accuracy should increase as well.







[1] Learning to Prompt for Continual Learning, CVPR 2022

**Questions:**

See Weaknesses

**Limitations:**

The authors addressed the limitations.

---

> ### Author Rebuttal · Authors · 2024-08-07
>
> Thank you very much for the constructive comments and questions. We are grateful for the positive comments on our design and empirical justification. Please see our detailed one-by-one responses below.
>
> >**Weaknesses #1:** TPP is similar to L2P [Ref2].
>
> Although our TPP and L2P both adopt prompting for class-incremental learning, several major differences highlight our novelty.
> - **Data:** L2P focuses on continual learning in the vision domain where each instance is independent while TPP addresses the graph continual learning where different nodes are connected by edges and not independent. The complex connections make graph continual learning more challenging.
> - **Prompt learning:** L2P follows the key-valued query mechanism to select prompts for inputs and the selected prompts are concatenated with input embedding. In contrast, our prompts operate on raw attributes of nodes and modify them with weighted combinations of prompts. Moreover, L2P employs ViT-B/16 as the backbone, which contains excessive parameters and is well-trained on large-scale auxiliary datasets. However, there are no universal backbones for graph learning. To address this, we learn the GNN backbone based on the first task via graph contrastive learning due to its ability to obtain transferable models across graphs.
> - **Forgetting:** Most importantly, L2P does not have task prediction (TP), but TP is the critical mechanism that gives TPP superior performances. By performing TP and learning task-specific prompts, TPP is fully rehearsal-free and forgetting-free. However, the forgetting problem is still a major issue in L2P.
>
> >**Weaknesses #2.1** Why the GCN-based OOD detector (OODCIL) underperforms compared to the proposed TP and baselines.
>
> We agree that Laplacian smoothing (LS) is similar to GCN. However, the proposed Laplacian smoothing-based task prediction (TP) is significantly different from the OOD detector-based method (OODCIL). First, we employ LS to construct the task prototypes without any training to profile each task. By contrast, OODCIL needs to fully train an OOD detector to discriminate the current task and OOD data of other tasks. Second, given a test sample, TP explicitly predicts the task ID by finding the most similar prototype while OODCIL relies on the OOD score to predict the task probability. As a result, the task prediction of OODCIL heavily relies on the performance of the detector. In our experiments, the OOD detector is trained with the current task and OOD data from other tasks. As a result, the OOD detector cannot effectively output the correct OOD score for a test sample, leading to poor performance. Designing more advanced OODCIL may improve the performance, but it is out-of-scope for this paper.
>
> As suggested, we conducted the experiments by changing the task prediction method in OODCIL and TPP and reported the results in the following table. When using our TP method for task prediction, the performance of OODCIL is significantly improved, similar to the performance of our full TPP method (Graph Prompting + TP). On the other hand, the performance of TPP drops significantly if the OOD-based method is used in TPP for task prediction, showing that accurate task prediction plays a critical role in GCIL’s impressive performance.
> ```
> Table A1: The average performance of OODCIL and TPP with different task prediction methods.
> ```
> |Method|Task Prediction|CoraFull|Arxiv|Reddit|Products|
> |---|---|---|---|---|---|
> |OODCIL|OOD|71.3|19.3|79.3|41.6|
> |OODCIL|TP|94.6|84.6|99.6|95.1|
> |Graph Prompting|OOD|1.5|4.3|6.0|8.0|
> |Graph Prompting|TP|93.4|85.4|99.5|94.0|
>
> Note that Graph Prompting+TP performs similarly to OOD-CIL+TP. It highlights the importance of our graph prompting learning from another perspective. This is because OODCIL has much more learnable parameters than our Graph Prompting as OODCIL trains a separate GNN for each task whereas Graph Prompting trains only a small GNN once and then learns small prompts for each task using a frozen GNN. Furthermore, OODCIL requires data replay, while our graph prompting is replay-free.
>
>
>
> >**Weaknesses #2.2:** Accuracy of task prediction with different task formulations.
>
> The following table reports the accuracy of the proposed TP with other task formulations, i.e., descending and random orders, demonstrating that the proposed TP can accurately predict the task IDs in terms of all formulations.
> ```
> Table A2: The accuracy of task prediction with other task formulations.
> ```
> |Task Formulation|CoraFull|Arxiv|Reddit|Products|
> |---|---|---|---|---|
> |Descending|100|100|100|100|
> |Random|100|100|100|100|
>
>
> >**Weaknesses #3:** Absence of source code.
>
> The source code will be released upon acceptance.
>
> >**Weaknesses #4:** Accuracy of TP with varying steps of Laplacian smoothing.
>
> In Theorems 1 and 2, we assume a sufficiently large number of Laplacian smoothing (LS) steps for task prediction. Specifically, for different graph tasks $\mathcal{G}_i$ and $\mathcal{G}_j$ consisting of different graph data, larger steps result in the train and test prototypes of the same task being the same (Theorem 1) and ensure the prototypes of different graphs are distinct (Theorem 2) simultaneously. In other words, the over-smoothing problem of LS is not the problem that we aim to address but it is the property that we utilize for accurate task prediction. We report the accuracy of task prediction with varying LS steps in the following table. We can see that the task IDs can be perfectly predicted even with one step of LS. This is attributed to the discriminability among node attributes in different tasks, e.g., task identification can be well predicted even without LS.
>
> ```
> Table A3. The accuracy of task prediction with different Laplacian smoothing steps.
> ```
> |Steps|0|1|3|5|
> |---|---|---|---|---|
> |CoraFull|97.14|100|100|100|
> |Arixv |100 |100|100|100|
> |Reddit|100|100|100|100|
> |Products|95.65|100|100|100|
>
>
> **References**
> - [Ref2] Learning to Prompt for Continual Learning, CVPR 2022

---

> > ### Comment · Reviewer_j8FV · 2024-08-10
> >
> > I appreciate the thorough response of the authors. However, I'm still suspicious about your task prediction performance.
> >
> > Specifically, the authors claimed that "given a test sample, TP explicitly predicts the task ID by finding the most similar prototype while OODCIL relies on the OOD score to predict the task probability. As a result, the task prediction of OODCIL heavily relies on the performance of the detector." However, this explanation does not fully address my concerns. The proposed TP is essentially a neighborhood normalized aggregation using their raw node features. On the other hand, the GCN-based OOD detector includes a GCN encoder that functions similarly through neighborhood aggregation. Moreover, the GCN encoder is trained specifically to discriminate between the current task and other tasks—a step that the proposed TP lacks. Therefore, it remains unclear to me why TP significantly outperforms the GCN-based OOD detectors by such a wide margin. The results presented in the third row of Table A1 (i.e., GraphPrompt + OOD) seem particularly questionable, as the performance is inexplicably low (nearly equivalent to that of a random classifier). The authors should clarify and convince readers of the specific factors that enable TP to outperform a seemingly similar approach (GCN-based OOD detector) so dramatically.
> >
> > My skepticism is further heightened by the results in Table A3. For the Arxiv and Reddit datasets, the performance reaches "100%" accuracy without any Laplacian steps, which implies that simply utilizing node features can perfectly discriminate between tasks. In other words, task prediction appears to be an exceptionally easy problem that can be fully resolved by merely calculating the similarity of raw node features. If that is the case, why does the GCN-based OOD detector struggle with such an apparently straightforward problem?
> >
> > In summary, I still have significant concerns regarding the perfect performance reported for the proposed TP:
> >
> > What specific factors allow the proposed TP to significantly outperform the GCN-based OOD detector?
> > How does the method achieve perfect prediction performance by merely using raw node features and their prototypes for similarity search?
> > Given these concerns, I strongly urge the authors to provide the source code, as it is the most effective way to address these issues.
> > I am very open to discussing these concerns further with the authors.
> >
> > Best regards,
> > Reviewer j8FV

---

> ### Author Response · Authors · 2024-08-12
>
> Thank you so much for the insightful comment.
>
> The source code of TPP is provided at https://anonymous.4open.science/r/TPP-1B07/README.md, where we provide the implementations of TPP and OODCIL in "Baselines/tpp\_model" and "Baselines/ood\_model" respectively.
>
> To explain why the proposed prototype-based methods can achieve higher task prediction accuracy than the OOD-based method, we'd like to clarify the key differences between them despite they share similar neighborhood aggregation strategies.
>
> Given a sequence of connected graphs (tasks) $(\mathcal{G}^1, \ldots, \mathcal{G}^T)$, where each task contains a set of unique $C$ classes of graph data, TPP constructs task prototypes for each task at its training stage, denoted as $\mathcal{P} = (\mathbf{p}^1,\ldots,\mathbf{p}^T)$. During inference, the prototype $\mathbf{p}^{\text{test}}$ for the test task is constructed, and task prediction is performed by identifying the most similar prototype in $\mathcal{P}$. Note that **this process in TPP does NOT involve any training and data reply**. Despite the simplicity, the prototype-based task prediction can achieve surprisingly good performance. This is attributed to the discrimination of the graph structure and node attributes between different tasks, as shown in Figure 3 in the paper.
>
> **Different from our training-free and reply-free method, OODCIL requires training an OOD detector for each task using data from the current task as in-distribution (ID) data and the rehearsal data from the other tasks as OOD data, and it then utilizes the OOD score to perform task prediction**. This means there are $T$ OOD detectors after learning $T$ tasks. During inference, a test graph is fed into all the $T$ detectors that obtain an OOD score for each task, and the test graph is predicted to belong to the task with the lowest OOD score. Specifically, let $f_o^t(\cdot)$ be an OOD detector trained for task $t$ and there will be $T$ detectors: $(\{ f_o^1(\cdot), \ldots, f_o^T(\cdot)\})$ after sequentially learning $T$ tasks. Then, for a test graph $\mathcal{G}^{\text{test}}$, ideally, the learned OOD detector $f_o^t(\cdot)$ should yield the lowest OOD score if $\mathcal{G}^{\text{test}}$ comes from task $t$ and output a high OOD score if otherwise. To endow OOD detector $f_o^t(\cdot)$ with such an ability, the ideal case is that we have ID data from task $t$ and the OOD data from all other tasks. However, due to the sequential emergence of graph tasks and the restriction of access to previous tasks, we treat the current graph at task $t$ as ID data and construct the OOD data by a data replay approach (i.e., sampling subgraphs from all previous $t-1$ tasks) in our experiments. The detector $f_o^t(\cdot)$ is then optimized to perform a $(C+1)$-way classification, where the first $C$ entries of the classification probabilities are for ID classes at task $t$ and the $(C+1)$-th probability output is used to define the OOD score.
>
> However, the OOD detector $f_o^t(\cdot)$ can get only limited access to the graph data from all $(t-1)$ previous tasks (i.e., having access to the replay data only). Moreover, when training $f_o^t(\cdot)$, we also do not have any access to graph data of unseen task $j\in(t, T]$. Thus, *due to the lack of sufficient training samples for seen tasks and the absence of the samples of unseen task $j$*, the trained $f_o^t(\cdot)$ can yield a lower score for the task $j$ than the OOD score yielded by $f_o^j(\cdot)$. This means that $f_o^j(\cdot)$ can often produce a lower OOD score for task $j$ than for the other tasks, but the lowest OOD score yielded by  $f_o^j(\cdot)$ is still smaller than the OOD score yielded by the other detectors for the same task $j$, leading to incorrect task prediction. For example, in Table A4 below, the OOD score for task 3 yielded by $f_o^1(\cdot)$ is lower than that yielded by $f_o^3(\cdot)$ (note that task $3$ has the lowest OOD score among the OOD scores yielded by $f_o^3(\cdot)$), and the OOD scores for task 4 and task 5 yielded by $f_o^1(\cdot)$ is lower than that yielded by $f_o^4(\cdot)$ and $f_o^5(\cdot)$ respectively. As a result, the OOD scores yielded by OOD detectors trained at earlier tasks are generally very low, e.g., the OOD scores in columns "$f_o^1(\cdot)$" and "$f_o^2(\cdot)$" in Table A4, leading to incorrect prediction of the tasks 3, 4 and 5 to be task 1.
>
> ```
> Table A4. OOD scores of each test task which are yielded by all OOD detectors on the Arxiv dataset with 5 tasks. The test graph is predicted to be the task ID whose OOD detector yields the smallest OOD score.
> ```
>
> |Test Graph|$f_o^1$|$f_o^2$|$f_o^3$|$f_o^4$|$f_o^5$|
> |---|---|---|---|---|---|
> |Task 1|**0.08**|0.92|0.90|0.89|0.94|
> |Task 2|0.16|**0.07**|0.86|0.87|0.89|
> |Task 3|**0.13**|0.31|0.17|0.92|0.85|
> |Task 4|**0.16**|0.42|0.82|0.34|0.96|
> |Task 5|**0.10**|0.37|0.48|0.93|0.29|

---

> > ### Author Response · Authors · 2024-08-12
> >
> > These two adverse effects lead to inaccurate task prediction of the OOD score-based approach, much less accurate than the proposed prototype-based approach (please see Table A5). This is mainly due to the fact that i) its training has a strong reliance on sufficient OOD data from the tasks other than the current task and ii) this training performance (or the access to other task data) is largely restricted due to the CIL nature. Designing an effective OOD detector for graph data has not been explored in the literature. It is challenging due to the limited access to OOD data. If we continuously update the OOD detector, we will need to handle catastrophic forgetting problem with the OOD detector and its interference with the CIL classifiers as well. So, in this work, we implement a simple approach for OOD detectors, but we agree that it is an important problem for GCIL.
> >
> > ```
> > Table A5. Task prediction accuracy of three different methods.
> > ```
> > |Method|CoraFull|Arxiv|Reddit|Products|
> > |---|---|---|---|---|
> > |Laplacian Smoothing|100|100|100|100|
> > |Node Attributes|97.14|100|100|95.65|
> > |OOD|62.86|10.00|80.00|78.26|
> >
> > Besides, the reason for GraphPrompt+OOD barely working is that GraphPrompt+OOD has significantly fewer learnable parameters than OODCIL, which leads to less accurate node classification per task. Combined with less inaccurate task prediction by the OOD detection method, the overall CIL performance is remarkably degraded.
> >
> > We very much hope the source code and our responses have addressed your concerns. We're more than happy to take any further questions if otherwise. Please kindly advise. Thank you very much!
> >
> > Best regards,
> >
> > Authors of Paper 8497

---

> > > ### Author Response · Authors · 2024-08-13
> > >
> > > Dear Reviewer j8FV,
> > >
> > > We have provided detailed replies and newly added empirical results to address your concerns on why the OOD detection method are not as effective as our proposed prototype-based task prediction method. Per your request, we have also released our codes through an anonymous GitHub repository. Could you please kindly check whether they are helpful for answering your questions? We're ready to take any further questions you might have.

---

> > > > ### Comment · Reviewer_j8FV · 2024-08-14
> > > >
> > > > I have carefully reviewed the code and verified the 100% accuracy of task ID prediction. I appreciate your effort, and most of my concerns have been addressed. As a result, I will be raising my score.

---

> > > > > ### Author Response · Authors · 2024-08-14
> > > > >
> > > > > Dear Reviewer j8FV,
> > > > >
> > > > > Thank you so much for your time and great support on verifying our code and the task ID prediction accuracy, and for the increase to an acceptance score. We're very pleased to know that our response has addressed most of your concerns. Please kindly let us know if there are any further questions.

---

### Official Review · Reviewer_odiT · 2024-07-12

**Soundness:** 3
**Presentation:** 2
**Contribution:** 3
**Rating:** 5
**Confidence:** 3

**Summary:**

This paper studies the graph class-incremental learning problem with unknown task identities. Specifically, the unknown task identity is the key challenge, and this work proposes a Laplacian smoothing-based graph task profiling approach that is theoretically justified capable of predicting the task identities.

Besides, the forgetting problem is alleviated through a graph prompting approach. This approach learns a graph prompt for each task, such that the classification models for different tasks are separated.

The experiments are conducted on four datasets.

**Strengths:**

The proposed method can achieve 100% task identification accuracy.

Theoretical analysis is also provided for the proposed method.

The proposed method is compared against multiple SOTA baselines and obtains consistent performance improvement.

**Weaknesses:**

The experimental setup is not introduced with details.

The introduction on the background is not clear enough. For example, what is the relationship between identifying the tasks and overcoming the forgetting problem. Is recognizing the tasks correctly enough for avoiding the forgetting problem.

**Questions:**

1. How are the datasets split into different tasks? Will different splitting strategy affect the performance?

2. What is the relationship between identifying the tasks and overcoming the forgetting problem. Is recognizing the tasks correctly enough for avoiding the forgetting problem.

**Limitations:**

As mentioned by the authors, the main limitation of the work is the limited representative capacity and generalizability of the GNN backbone model in prompting.

---

> ### Author Rebuttal · Authors · 2024-08-07
>
> Thank you very much for the constructive comments and questions. We are grateful for the positive comments on our design and empirical justification. Please see our detailed one-by-one responses below.
>
> >**Weaknesses #1/Questions #1:** Not clear experimental setup and the performance of TPP with different task formulations.
>
> Please refer to our reply in **Global Response to Shared Concerns** in the overall Author Rebuttal section above for this concern.
>
>
> >**Weaknesses #2/Questions #2:** What is the relationship between identifying the tasks and overcoming the forgetting problem?
>
> In class-incremental learning, the absence of task ID or identification information requires the test instance to be classified into one of all learned classes, leading to the challenge of inter-task class separation. This can exacerbate the forgetting problem in class-incremental learning. By accurately identifying the task ID, the classification is constrained within the task and the forgetting problem can be largely alleviated as shown in Table 2 in the paper. However, it cannot fully overcome the forgetting problem due to the knowledge interference between tasks when training a single model. In this paper, we further address this issue by learning and storing task-specific knowledge in graph prompts, resulting in the proposed method being forgetting-free.

---

### Official Review · Reviewer_4w4P · 2024-07-14

**Soundness:** 2
**Presentation:** 3
**Contribution:** 2
**Rating:** 5
**Confidence:** 4

**Summary:**

The paper proposes a Replay-and-Forget-Free Graph Class-Incremental Learning (GCIL) approach called Task Profiling and Prompting (TPP). This method addresses the challenges of class-incremental learning in graph tasks without relying on task identifiers during inference. By using Laplacian smoothing-based task profiling for accurate task ID prediction and a novel graph prompting approach, TPP eliminates catastrophic forgetting and improves classification accuracy across multiple tasks.

**Strengths:**

- The problem of obtaining a GCIL model being both replay-free and forget-free is interesting.

- The presented empirical results are good.

- The theoretical analysis is interesting.

**Weaknesses:**

- Although CIL problem is interesting, the problem definition in Section 3.1 requires some clarification. Are there any real setting that a sequence of connected graphs (tasks) appear during training, these graphs may be the same but the target tasks are different?
Besides, during testing, why a GCIL model is required to classify a test instance into one of all the T × C classes? Should it be firstly identify its tasks (1 of T), then assign to C classes?


- Likewise, the datasets used in experiments, have been applied for routine scenarios. Why they should be considered into GCIL setting?

**Questions:**

- Why "Despite being only learned on G1, f(·) can effectively adapt to all subsequent tasks with graph prompts" as written on line 228-229? Will this lead to high fluctuation if we choose different G1? Could the authors provide some empirical results on that?


Please also check my comments above.

---

> ### Author Rebuttal · Authors · 2024-08-07
>
> Thank you very much for the constructive comments and questions. We are grateful for the positive comments on our design and empirical justification. Please see our detailed one-by-one responses below.
>
> >**Weaknesses #1:** The problem definition requires some clarification.
>
> In the **Global Response to Shared Concerns**, we clarify the task formulation in our experiments. In our setting, we follow the widely-used GCIL setting in [Ref1], where a new graph $\mathcal{G}^t$ that connects with previously occurred graphs emerges with a new set of target classes at each time step $t$. The setting may be applied to various real-world applications. For example, in a bank transaction network where each account is a node and the edge can be built when there are transactions between two accounts, the newly emerging graph can be formed by new accounts/transactions with new types (classes) of transactions.
>
> During inference, GCIL aims to classify a test instance into one of all the learned classes (i.e., $T\times C$ classes). Most existing GCIL works directly perform the classification without identifying the task ID of test instances, so their class set for each test node includes $T\times C$ classes. An alternative approach, as suggested by you, is to first identify its task ID, and then perform C-way classification. This is the approach we take in our proposed TPP. This type of methods requires accurate task prediction to guarantee good CIL performance. Inspired by this, we propose the TPP method in this paper, in which the proposed task profiling module shows remarkable task prediction accuracy.
>
> >**Weaknesses #2:** Why these datasets should be considered into the GCIL setting?
>
> Following [Ref1], we employ CoraFull, Arxiv, Reddit, and Products as the benchmark datasets for GCIL and follow the same task formulation. This ensures fair comparisons to all methods. The reason for choosing these datasets for GCIL can be attributed that they contain multiple classes so that a diverse set of tasks can be constructed to evaluate the performance of GCIL methods.
>
> >**Question #1:** The performance of TPP with different task formulations.
>
> Our method can achieve stable CIL performance with different starting and subsequent graph compositions. Please refer to our **Global Response to Shared Concerns** in the overall Author Rebuttal section above for newly added empirical results that justify this ability.
>
> **References**
>
> - [Ref1] CGLB: Benchmark Tasks for Continual Graph Learning. NeurIPS 2022 Datasets and Benchmarks Track.

---

### Author Rebuttal · Authors · 2024-08-03

Dear all reviewers,

Thank you very much for the time and effort in reviewing our paper, and for the constructive and positive comments. Our rebuttal consists of two parts: **Global Response** where we address the shared concerns from two or more reviewers and **Individual Response** where we provide a detailed one-to-one response to address your questions/concerns individually.

>**Global Response to Shared Concerns**: The performance of TPP with different task formulations.

For the task formulation in our experiments, we set each task to contain two different classes of nodes and follow the commonly used task formulation strategy in [Ref1] to have fair comparisons with the baselines. Specifically, given a graph dataset with many classes, we split these classes into different tasks in numerically ascending order of the original classes, i.e., classes 0 and 1 form the first task, classes 2 and 3 form the second task, and so on. To evaluate the performance of TPP with different task formulations, we further perform the class splitting in two other manners, including numerically descending and random ordering of the two classes per task. In the following table, we report the average performance of the TPP and the Oracle Model with different task formulations.
```
Table A1. Results of average performance of TPP and Oracle Model on datasets with various task formulations.
```
|Task Formulation | Method | CoraFull | Arxiv | Reddit | Prodcuts|
|----|----|----|----|----|----|
|Ascending Order (Reported)| TPP| 93.4|85.4|99.5|94.0|
|Ascending Order (Reported)| Oracle Model|95.5|90.3|99.5|95.3|
|Descending Order|TPP|94.5|85.9|99.4|93.9|
|Descending Order|Oracle Model|96.1|91.6|99.5|94.7|
|Random Order|TPP|94.8|86.9|99.5|85.9|
|Random Order|Oracle Model|95.3|91.3|99.7|86.8

From the table, we observe that the proposed TPP method can still achieve comparable performance to the Oracle Model with different task formulations, highlighting the robustness and effectiveness of TPP w.r.t. the formulation of individual tasks. Note that the performances of TPP and Oracle Model both drop on Products with random task formulation. This is attributed to the heavily imbalanced class distribution of Products and the performance is evaluated by the balanced classification accuracy. Specifically, for Products, some classes contain hundreds of thousands of nodes while the number of nodes in some classes is less than 100. The ascending and descending task formulations have a relatively balanced class distribution for each task. However, the random task formulation results in some tasks with heavily imbalanced class distribution. To address this problem, de-biased learning is required. We leave it for future research.

Please also note that TPP learns the GNN backbone only on the first task and is frozen during the subsequent prompt learning. Different task formulations result in the GNN backbone being learned with different first tasks. The above results reveal that the proposed graph prompting enables the learned GNN backbone to effectively adapt to all subsequent tasks despite the backbone being learned on different initial datasets.


As for **Individual Response**, we have provided a detailed one-by-one response to answer/address your questions/concerns after the post of your review.

We very much hope our responses have cleared the confusion, and addressed your concerns. We're more than happy to take any further questions if otherwise. Please kindly advise!

Best regards,

Authors of Paper 8497

**References**

- [Ref1] CGLB: Benchmark Tasks for Continual Graph Learning. NeurIPS 2022 Datasets and Benchmarks Track.

---

### Decision · Program_Chairs · 2024-09-25

**Decision:**

Accept (poster)

**Comment:**

This paper investigates a graph class-incremental learning problem with unknown task identities. The authors designed a novel graph task profiling and prompting (TPP) approach. They also provided theoretical analysis to show that a simple Laplacian smoothing-based graph task profiling approach can achieve accurate graph task ID prediction. Extensive experimental results are reported and discussed. All the reviewers recognized the technical contributions of this work. Also, during the post-rebuttal discussion, reviewers agreed that their previous concerns have been addressed in the rebuttal. The authors are highly encouraged to further improve their final version by incorporating the comments and suggestions from reviewers.